# Widespread Decline in Terrestrial Water Storage and Its Link to Teleconnections across Asia and Eastern Europe

Xianfeng Liu[1,2], Xiaoming Feng[1], Philippe Ciais[3], Bojie Fu[1]

[1]State Key Laboratory of Urban and Regional Ecology, Research Center for Eco-Environmental Sciences, Chinese Academy of Sciences, Beijing 100085, China
[2]School of Geography and Tourist, Shaanxi Normal University, Xi'an 710119, China
[3]Laboratoire des Sciences du Climat et de l'Environnement, CEA-CNRS-UVSQ, Gif-sur-Yvette, France

*Correspondence to*: Xiaoming Feng (fengxm@rcees.ac.cn)

**Abstract.** Recent global changes in terrestrial water storage (TWS) and associated freshwater availability raise major concerns over the sustainability of global water resources. However, our knowledge regarding the long-term trends in TWS and its components is still not well documented. In this study, we characterize the spatiotemporal variations in TWS and its components over the Asian and Eastern European regions from April 2002 to June 2017 based on Gravity Recovery and Climate Experiment (GRACE) satellite observations, land surface model simulations, and precipitation observations. The connections of TWS and global major teleconnections (TCs) are also discussed. The results indicate a widespread decline in TWS during 2002–2017, and five hotspots of TWS negative trends were identified with trends between -8.94 mm yr$^{-1}$ and -21.79 mm yr$^{-1}$. TWS partitioning suggests that these negative trends are primarily attributed to the intensive over-extraction of groundwater and warm-induced surface water loss, but the contributions of each hydrological component vary among hotspots. The results also indicate that the El Niño-Southern Oscillation, Arctic Oscillation and North Atlantic Oscillation are the three largest, dominant factors controlling the variations in TWS through the covariability effect on climate variables. However, seasonal results suggest a divergent response of hydrological components to TCs among seasons and hotspots. Our findings provide insights into changes in TWS and its components over the Asian and Eastern European regions, where there is a growing demand for food grains and water supplies.

## 1 Introduction

Terrestrial water storage (TWS), a fundamental component of terrestrial hydrological cycle (Tang et al., 2010), represents the total water stored above and below a land surface (Syed et al., 2008). TWS is composed of surface water (SW), including lakes, snow water equivalent, canopy water and glaciers, soil moisture (SM), and groundwater (GW) storage (Ni et al., 2018; Cao et al., 2019). Changes in TWS are strongly affected by climate change, e.g., drought, floods, prolonged high temperatures, and anthropogenic activities, e.g., abstraction-driven groundwater depletion. Recent TWS information has raised worldwide concerns because of its association with freshwater availability and concerns about the sustainability of

global water resources (Creutzfeldt et al., 2012; Meng et al., 2019). Accurate monitoring and quantification of TWS is therefore critical for sustainable water resources management.

Gravity Recovery and Climate Experiment (GRACE) satellites measured global TWS changes from April 2002 to June 2017 (Reager et al., 2009), which provided hydrologists with practical insights at regional and global scales in comparison to in situ measurements (Zhang et al., 2015; Cao et al., 2019). With GRACE data, previous literature has mostly focused on the

TWS changes at the basin (Zhang et al., 2015; Shamsudduha et al., 2017; Yang et al., 2017), regional (Rodell et al., 2009; Long et al., 2013; Creutzfeldt et al., 2015; Ndehedehe et al., 2017) or continental scale (Syed et al., 2008; Rakovec et al., 2016; Yi et al., 2016; Ni et al., 2018). For instance, Rakovec et al. (2016) analyzed the TWS anomaly using GRACE in 400 European river basins. GRACE data also contributed to the exploration of hydrological storage changes, e.g., glacial mass loss (Jacob et al., 2012; Yi et al., 2014; Brun et al., 2017; Huss et al., 2018), lake level and extent changes (Zhang et al., 2013;

Zhang et al., 2017), and groundwater depletion (Rodell et al., 2009; Wada et al., 2010; Döll et al., 2014; Long et al., 2016; Feng et al., 2018; Tangdamrongsub, et al., 2018). However, few studies have focused on the contributions of hydrological components to TWS variability at a large scale, particularly in water-limited and densely populated regions (Tapley et al., 2019). Two recent global scale studies substantially improved our knowledge by identifying 34 hotspots of TWS changes during 2002–2016 (Rodell et al., 2018) and the changes in global endorheic basin water storage (Wang et al., 2018).

The Asian and Eastern European regions, home to half of the world's population and 50% of its arid/semiarid climate areas, are undergoing intensive water exploitation to agriculture and domestic water needs (Huang et al., 2016) (Figure 1). Most of the countries located within the borders of Asia and Eastern Europe are experiencing water resource shortages caused by low annual precipitation (less than 400 mm yr$^{-1}$); when the area of these countries are combined, they consequently comprise the largest amount of irrigated land in the world (Rodell et al., 2009). The increasing demand for freshwater and the limited

knowledge on the available water resources in this region have become the key challenge to achieving sustainability in these areas (Feng et al., 2013). Therefore, knowledge on the TWS trend and its hydrological components is important for the sustainable development of water resources and food supplies in this region.

The large-scale mode of teleconnection (TC) is an overwhelming factor in regional water resources, modulating the location and strength of storm tracks and fluxes of heat, moisture, and momentum. For example, prominent teleconnection patterns

such as El Niño-Southern Oscillation (ENSO) show that El Niño years are related to reduced precipitation, continental freshwater discharge, and evapotranspiration over many land areas; therefore, TWS variability occurs over many land areas (Gonsamo et al., 2016). Many studies have attempted to address the possible causes of TWS changes by connecting TWS with TC (Phillips et al., 2012; Ndehedehe et al., 2017; Ni et al., 2018; Forootan et al., 2019). However, these studies focused primarily on the effect of ENSO on TWS. Notably, many other global and regional climate TCs have also influenced the

changes in TWS; these TCs, however, have been less extensively documented, which consequently limits our understanding of a comprehensive TWS-TC correlation. Therefore, knowledge of the influence of multiple TCs on TWS is critical for improving our understanding and proper management of water resources (Phillips et al., 2012; Ndehedehe et al., 2017).

In this study, we conducted a comprehensive analysis of the spatiotemporal variations in TWS across the Asian and Eastern European regions and address the contributions of each hydrological component and connection with TCs using multisource data. First, we calculated the de-seasonalized trend and analyzed the spatiotemporal variations in TWS across Asia and Eastern Europe. Then, we partitioned the components of TWS into SW, SM, and groundwater by using GRACE, the Noah land surface model, and lakes and glacial observation data. Finally, we calculated the cross-correlation coefficients between TCs and the detrended and de-seasonalized TWS time series. We aimed to explore 1) the spatial pattern of long-term trends in TWS, 2) the contributions of water components to TWS variations among regions, and 3) the role of TCs in the changes in TWS and its components within the Asian and Eastern European regions.

## 2 Materials and Methods

### 2.1 Study area

The Asian and Eastern European regions, with arid and semiarid land comprising 54% of its total area, are located between latitudes 6°S and 56°N and longitudes 4°E and 109°E (Figure 1a). These regions are the most densely populated regions in the world, sustaining nearly half of the global population (Gridded Population of the World: GPW, v4), and contain some of the largest and most intensively irrigated lands of the world (Figure 1b). Freshwater availability in these water-limited regions is essential to food and water security and, hence, sustainable economics. However, the amount of available freshwater in these regions is highly dependent on precipitation and temperature (Wang, 2018); consequently, these factors are influenced intensively by the Northern Hemisphere atmospheric circulation patterns and the coupled ocean-atmosphere patterns (i.e., teleconnections). Therefore, spatially explicit analyses of the impacts of teleconnections on freshwater availability in these regions can provide a simple framework for understanding the complex response of freshwater availability to global climate change.

[Please Insert Figure 1 Here]

### 2.2 Data

GRACE satellite measures the vertical terrestrial water storage from the land surface to the deepest aquifers and can be used to monitor spatiotemporal variability in terrestrial water storage anomalies (Scanlon et al., 2016). The advanced mass concentration (mascon) approach contains a much higher signal-to-noise ratio in TWS retrieval than the traditional global spherical harmonics (SH) technique because of reduced leakage errors (Scanlon et al., 2018, 2019). Notably, the GRACE mascon solutions derived from the Jet Propulsion Laboratory (hereafter JPL-M) and the Center for Space Research (CSR-M) represent two fundamentally different approaches to applying constraints. The constraints applied in the JPL-M processing are based on both GRACE data and geophysical models for mass changes over land, oceans, ice, inland seas, earthquake areas, and areas affected by glacial isostatic adjustments (Watkins et al., 2015), whereas CSR-M constraints are based solely on GRACE data (Save et al., 2016). Meanwhile, the JPL-M solution has the unique characteristic that each 3°×3°

(approximately to the native resolution of GRACE data) mascon element is relatively uncorrelated with the neighboring
mascon elements (Rodell, 2018). In this study, both the JPL-M and the CSR-M datasets were used to detect TWS changes
across the Asian and Eastern European regions with spatial resolutions of 0.5° × 0.5° for the period between April 2002 and
June 2017, for a total of 183 solutions. The missing data for the duration of 20 months in the original time series were filled
by the linear interpolation method (Long et al., 2015; Yang et al., 2017). The GRACE anomalies reported in these mascon
solutions are relative to a 2004–2009 mean baseline (Scanlon et al., 2016). For details on the data processing, please refer to
Watkins et al. (2015) and for the mascon solutions, Save et al. (2016).

The Global Land Data Assimilation System (GLDAS) data between April 2002 and June 2017 was used to partition the
GRACE-observed TWS changes into SW (snow water equivalent, canopy water, lakes and glaciers), SM and groundwater.
The monthly data products from the GLDAS version 2.1 Noah model contain 36 variables, including canopy water storage,
snow water equivalent and SM data. Noah has a total of 4 layers of SM thickness: 0–10, 10–40, 40–100, and 100–200 cm.
To compute the GLDAS TWS, the SM in all layers, the snow water equivalent, and canopy SW are summed. The summed
GLDAS TWS is comparable to GRACE TWS over land (Rodell et al., 2004), and, notably, the GLDAS version used here
does not include groundwater and separate SW components (such as rivers and lakes). Therefore, deviations from the
GRACE total water storage changes can be expected. A comparison between GRACE and GLDAS is shown in Figure S1.
The native spatial resolution of the GLDAS dataset is 0.25° × 0.25°; we resampled these data to a 0.5° × 0.5° spatial
resolution using the nearest neighbor interpolation method prior to the analysis. The sea level (Caspian Sea, East Aral Sea,
West Aral Sea, and North Aral Sea) data derived from multimission altimeter observations were obtained from the Database
for Hydrological Time Series of Inland Water (Schwatke et al., 2015) and Hydroweb (Crétaux et al., 2011). Glacier mass
change data (Tien Shan, Hindu Kush, Spiti Lahaul, East Nepal, Bhutan and Nyainqentanglha) are available from the
published literature (Brun et al., 2017; Wang et al., 2018). The groundwater counterpart was estimated by deducting the
estimated SW and SM changes from the GRACE-observed TWS change (Wang et al., 2018). Notably, interannual variations
in biomass are considerably below the detection limits of GRACE (Rodell et al., 2005; Rodell et al., 2009); therefore, the
variability in the SW counterpart mainly involves changes in lakes, snow, and ice in this study.

The term teleconnection may refer to patterns arising from the internal variability of the atmosphere as well as from the
coupling between the air and the ocean (Zhu et al., 2017). In this study, we analyze the TCs that dominate climate variability
in the Northern Hemisphere, including the Arctic Oscillation (AO), North Atlantic Oscillation (NAO), East Atlantic (EA),
East Atlantic/Western Russia (EAWR), Scandinavia (SCAND), Polar/Eurasia (polarEA), West Pacific (WP), Pacific/North
America (PNA); we also analyze four important atmosphere-ocean coupled variability patterns that influence global climate,
including the Indian Ocean Dipole (IOD), the Atlantic Multidecadal Oscillation (AMO), the Pacific Decadal Oscillation
(PDO), and ENSO (Zhu et al., 2017). The first 8 indices refer to Northern Hemisphere atmospheric circulation patterns.
These 8 indices were calculated for 20°N–90°N using a rotated principal component analysis (RPCA) of monthly mean
standardized 500-mb height anomalies (Barnston and Livezey, 1987). The IOD is defined by the difference in sea surface
temperature between two areas – a western pole in the western Indian Ocean (50~ 70° E, 10° S~ 10° N) and an eastern pole

in the eastern Indian Ocean (90~110° E, 10° S~ EQ). The IOD affects the climate of Asia and is an important contributor to rainfall variability in this region (Saji et al., 1999). The AMO and PDO indexes are defined as the leading principal component of the North Atlantic Ocean (0-65°N, 80°W-0°E) and the North Pacific Ocean (poleward to 20°N) monthly sea temperature variability, respectively (Enfield, 2001; Bond, 2000). ENSO is the most important coupled ocean-atmosphere phenomenon driving global climate variability. We adopted the monthly Multivariate ENSO Index (MEI) in this study, which considers the variability both in the atmosphere and in the ocean (Wolter and Timlin, 2011). All these indices were obtained from the Climate Prediction Center of the U.S. National Oceanic and Atmospheric Administration (Table 1).

**[Please Insert Table 1 Here]**

## 2.3 Methods

### 2.3.1 Time series decomposition

The original GRACE TWS signal is decomposed into long-term trends, seasonality signals, and residual components by implementing the Seasonal Decomposition of Time Series by Loess (STL) approach. The STL method is a robust method for time series decomposition, and the equation is as follows (Scanlon et al., 2016):

$$S_{total} = S_{long-term} + S_{seasonality} + Residuals,\tag{1}$$

where $S_{total}$ is the original signal, $S_{long-term}$ is the long-term trend, $S_{seasonality}$ is the seasonality signal and *Residual* is the residual component. An example is provided for a region in northwest India (Figure S2). For detailed principles and applications of STL, readers are encouraged to refer to Cleveland et al. (1990) and Bergmann et al. (2012).

### 2.3.2 Theil-Sen trend analysis

The de-seasonalized time series was used to calculate the linear trend of TWS and precipitation for the Asian and Eastern European regions from April 2002 to June 2017 using the Theil-Sen trend method. The advantage of the Theil-Sen trend analysis is that it is nonsensitive to outliers and therefore can be more accurate than a simple linear regression for skewed and heteroscedastic data (Sen, 1968). This method compares strongly against the least squares method, even for normally distributed data. The TWS trend, β, for a particular pixel is as follows:

$$\beta = \text{Median}\left(\frac{x_j - x_i}{j-i}\right), \forall j > i,\tag{2}$$

where $i$ and $j$ are the time series sequences and $x_i$ and $x_j$ are the TWS values at times $i$ and $j,$ respectively. When β>0, the TWS in the corresponding pixel reveals an increasing trend; when β<0, the TWS in this pixel reveals a decreasing trend. The significance of the trend is tested by using the Mann-Kendall statistical test (Kendall, 1955).

**2.3.3 Cross-correlation analysis**

Contemporaneous weather conditions impact TWS residuals and often show evident time lags. Therefore, in this study, we employ the cross-correlation method to explore the relationship between the TWS residual and teleconnection indices. The cross-correlation measures the similarity of the two time series datasets as a function of the displacement of one set relative to the other (Oppenheim et al., 2009). The cross-correlation is defined as follows:

$$\rho(\tau) = \frac{\sigma_{12}(\tau)}{\sqrt{\sigma_{11}\sigma_{22}}} \quad , \tag{3}$$

where $\sigma_{12}$ $(\tau)$ is the cross-covariance function of $x_1$ (t) leading the lagging $x_2$ (t), $\tau$ is the time lag, and $\sigma_{11}$ and $\sigma_{22}$ represent the auto-variances of $x_1$ (t) and $x_2$ (t), respectively. The value of $\rho$ $(\tau)$ lies between -1 and +1. Moreover, we focus on the current and historical influence of TCs on the TWS residual, and hence, we constrained the value of lag $\tau$ to range between 0 and 24 (Ni et al., 2018). Higher cross-correlation values indicate a stronger influence of a TC on a TWS residual and its

components. In this study, we calculated the cross-correlation at the residual time series to partially remove the influence of unaccounted factors on TWS changes; this is because we assumed that the trend was mainly induced by secular climate change or direct human impacts rather than interannual climate variability (Phillips et al., 2012; Wang et al., 2018). The methodology flow diagram in Figure 2 shows the detailed process for data processing and analysis.

**[Please Insert Figure 2 Here]**

**3 Results**

**3.1 Spatiotemporal changes in TWS**

Both GRACE-based solutions (JPL-M and CSR-M) show similar spatiotemporal pattern of changes in TWS (Figure 3 and Figure S3). Since the JPL-M solution has a lack of correlation between neighboring mascon elements in the retrieval, in this study we use JPL-M for trend analysis and mapping. JPL-M indicates that the Asian and Eastern European regions

experienced widespread declines in TWS during 2002–2017 (Figure 3a). Noticeably, the spatial regime of the TWS variation matches that of the precipitation trend, except for northwest India, areas north and east of the Caspian Sea, and the area north of Xinjiang in China (Figure 3b), thereby suggesting that variations in TWS in these regions are intertwined with human impacts. The North China Plain (Region 1), a vast agricultural region in China, has undergone a continuously negative trend in TWS (-8.94±3.91 mm yr$^{-1}$), although an increasing trend in precipitation is expected in the northern part of this region

(Figure 3b and Table S1). Another hotspot of TWS decline is located west of Urumqi in China's northwestern Xinjiang Province (Region 2), with a negative trend of -15.93±11.58 mm yr$^{-1}$. Rainfall in this region shows an increasing trend during the study period (Figure 3b). The most striking TWS deficit is in northwest India (Region 3), with a negative trend of -21.79±14.54 mm yr$^{-1}$. Two subcenters of TWS deficits (-11.74±8.11 mm yr$^{-1}$) are located in the border area of China, India, Bhutan, and Nepal (Region 4). The Middle East region witnessed the most widespread TWS depletion (-10.93±7.91 mm yr$^{-1}$)

in the study area (Region 5), and the Caspian Sea level showed a dramatically negative trend (-73.2 mm yr$^{-1}$) during the

GRACE era (Figure S4). The results also indicate that the decrease in precipitation is accompanied by an increase in temperature and evapotranspiration in region 5 (Figure 3b and Figure S5).

There are also several regions with increased TWS over the mid-high latitude, i.e., most regions of Russia and northeast China, coinciding with an increase in precipitation in these regions during the study period (Figure 3). Other hotspots with increased TWS in China during 2002–2017 are located in South China and the hinterlands of the Tibetan Plateau. In contrast to the sharp decline in TWS over northwest India, TWS in central and southern India exhibits an increasing trend during the GRACE era. The variability in southern monsoons and the associated increase in rainfall likely account for the positive trend in TWS (Rodell, 2018).

**[Please Insert Figure 3 Here]**

### 3.2 Influence of TC indices on TWS variability

Figure 4 shows the spatial distribution of the cross-correlation coefficients, illustrating the possible relationship of TCs with interannual variability in TWS. The results indicate that ENSO, AO, and NAO have a significant area of influence on TWS variability. Spatially, the pattern of correlation coefficients between TWS and ENSO is heterogeneous, with positive correlations occurring mostly in southeast China and boreal regions and negative correlations occurring in Southeast Asia, India, and eastern boreal regions. The second and third most dominant teleconnection modes are AO and NAO, respectively. AO mainly affects TWS variations across high-latitude regions through its impact on temperature variability, and NAO has a wider footprint that is scattered across the whole study area. Following the three dominant TCs, the positive effects of IOD are scattered throughout northwest India, southern Arabia, the European boreal region, northwest China, and the Yellow River basin, whereas the negative effects of IOD are mainly located in Southeast Asia. Other teleconnection modes typically have a smaller impact on TWS dynamics over the study area.

Proportions of time lags for different TCs are shown in Figure 4d. Nearly half of the area (49.14%) lags behind the TCs by up to 3 months, while the proportions of TWS variations lagging behind the TCs at 4–6 months and at 7–9 months are 20.27% and 12.28%, respectively. These time lags are mainly scattered in the mid-high-latitude region and the Yangtze River basin in China. Longer lags (10-18 months), accounting for 18.31%, are observed in parts of the Tibetan plateau, the Mongolia plateau, and the Middle East region. Notably, the spatial pattern of the dominant TC has only a limited extent with respect to their influence on climate conditions. The heterogeneous pattern highlighted the importance of focusing on the effect of multiple TCs on TWS rather than one teleconnection index.

**[Please Insert Figure 4 Here]**

### 3.3 Contributions of water storage components to TWS

The changes in TWS aggregate the contributions of different water storage components (Figure 5). Groundwater depletion (-8.68±2.89 mm yr$^{-1}$) dominates the contribution to TWS loss in region 1 compared to the other two components (-0.27±2.97 mm yr$^{-1}$ for SM, 0.02±0.11 mm yr$^{-1}$ for SW) during the study period. Similar results were observed in northwest India

(region 3 with -21.35 mm yr$^{-1}$), which contained the most extensive land irrigation area worldwide (Figure 1b). The contributions of the SW and soil water in the above two regions are extremely small and neglected. The rapid glacial mass loss in the Tien Shan Mountain range induced a 41% water loss (Jacob et al., 2012; Brun et al., 2017). The melt water and the increase in precipitation replenished the soil water, leading to an increase in soil water components (2.18±2.82 mm yr$^{-1}$) in northwest China (region 2). Notably, groundwater in region 2 contributed to more than half (-11.61±13.02 mm yr$^{-1}$) of the total water loss. In this region, both groundwater depletion and glacial melt may enhance evapotranspiration by pumping water from aquifers and mountains to the surface, contributing to a negative trend in TWS. The contributions of region 4 are 2.48±1.81 mm yr$^{-1}$, -3.22±7.24 mm yr$^{-1}$ and -11.00±10.43 mm yr$^{-1}$ for SW, SM, and groundwater, respectively, suggesting that groundwater withdrawal is the primary reason for TWS depletion. This region is also spatially coherent with precipitation deficits, as depicted in Figure 3b; this dynamic is especially prominent in the eastern part of this hotspot, thus accentuating the loss of TWS. The prominent SW loss (-3.83±1.68 mm yr$^{-1}$) in region 5 can be attributed to the prominent shrinkage in the Caspian Sea as demonstrated by a decrease in its water level elevation by -73.2 mm yr$^{-1}$, which is similar to the sea's -68 mm yr$^{-1}$ decline during 2002–2016 (Wang et al., 2018). The dramatic decline in the Caspian Sea level contributes to a third of the total TWS loss in this region. The large decrease in TWS in these areas can also be attributed to the heavy reliance on groundwater for irrigation and domestic needs due to the construction of dams upstream (Voss et al., 2013; Rodell et al., 2018). The slight decreasing trend in precipitation likely exacerbated the TWS depletion, as the change of TWS relied largely on precipitation in those water-limited regions. Our results suggest that TWS variations during 2002–2017 had different impacts on SW (lakes, biomass, snow, and ice), SM, and groundwater among the five hotspot regions.

**[Please Insert Figure 5 Here]**

### 3.4 Divergent response of water storage components to TCs

Our results indicate that the water storage components are simultaneously influenced by several teleconnections (Table S2). For instance, SM in region 2 significantly correlates with NAO, AO, EAWR, PNA, ENSO, IOD, EA, AMO, polarEA and PDO, with negative correlations for some indices and positive correlations with others. Moreover, the dominant teleconnection varies for different water storage components among the separate regions (Table S3). The changes in TWS and groundwater are generally less sensitive to TC signals compared to the surface and SM counterparts. A possible explanation may be that TWS is a synthesis signal, i.e., its trend will be offset by its components in different ways. The groundwater component intensively interferes with anthropogenic activities such as irrigation and domestic needs and groundwater withdrawal, which indicates a lower correlation with TCs.

Further seasonal analysis indicates that the response of water storage to TCs is seasonally different from one region to another (Figure 6). For example, TC signals have a dominant control on TWS and component variability in spring and summer for region 3 and region 1, respectively, whereas the signals control most of the changes in SM in region 5 in autumn and winter. Notably, although it has been thoroughly documented that the dramatic decline in TWS in northwest India can be attributed to the overexploitation of groundwater (Rodell et al., 2009), our seasonal response of water components to TCs

suggests that the SM in this region is highly correlated with spring ENSO signals (Figure 4, r = 0.77). These results highlight the importance of understanding the seasonal responses to TCs to improve predictions of changes in TWS and associated water storage components.

**[Please Insert Figure 6 Here]**

**4 Discussion**

**4.1 Comparison of TWS trends to existing studies**

We investigated the spatiotemporal trend of TWS and its components over Asia and Eastern Europe during 2002–2017. The spatial patterns and trends of TWS throughout the study area are consistent with those of previous studies (Humphrey et al., 2016; Scanlon et al., 2016). Our estimated trend (-8.94±3.91 mm yr$^{-1}$) of TWS in region 1 is similar to the results of previous

studies in this region (22±3 mm yr$^{-1}$ during 2003-2010) (Feng et al., 2013; Tangdamrongsub et al., 2018). The discrepancies may stem from the inconsistency of the study period and spatial domain. Due to a warm and dry long-term climate and intensive anthropogenic activities (agriculture, industry, and urbanization), the groundwater in region 1 has been overexploited since the 1970s, and more than 70% of the groundwater exploitation is used for regional irrigation (Wang et al., 2007). The groundwater loss rate in region 3 (-21.35 mm yr$^{-1}$) was also comparable with a previous study in region 3,

with approximately 40±10 mm yr$^{-1}$ from August 2002 to October 2008 (Rodell et al., 2009). As Indian agriculture leads the world in total irrigated land by consuming ~85% of the utilizable water resources (Salmon et al., 2015; Panda et al., 2016), a consensus has been reached that the dramatic decline in TWS is mainly due to the overexploitation (extraction exceeding recharge) of groundwater for irrigation (Shamsudduha et al., 2019). Although precipitation in region 3 shows an increasing trend during the GRACE period, the rapid depletion of TWS in northwest India induced by unsustainable consumption of

groundwater for irrigation and other anthropogenic uses has attracted worldwide attention because it is a major threat to India's sustainability (Rodell et al., 2009; Panda et al., 2016). Region 4 in our study is heavily irrigated (Figure 1); hence, intensive irrigation is likely to induce groundwater decline (-11.00±10.43 mm yr$^{-1}$). The increase in SW induced by melt water from mountains (Brun et al., 2017) was offset by the decrease in soil water, which may have been related to the decrease in precipitation (Figure 3b). For region 2, the rapid melting of the glaciers of Tien Shan Mountain accelerated the

losses of water resources, as the glacial meltwater will provide additional water that was lost to rivers or evaporation (Jacob et al., 2012). The negative trend in TWS indicates that water demand is larger than the supply in region 2, which can be attributed to both a continuous withdrawal of groundwater and extensive evaporation in the endorheic basin (Rodell et al., 2018). However, the increase in precipitation is expected to offset a certain portion of water losses in region 2. Previous studies also documented that the widespread decline in TWS in region 5 is attributed to an overreliance on groundwater for

domestic and agricultural needs stemming from human-made dams in addition to the considerable surface water loss (Voss et al., 2013; Joodaki et al., 2014; Rodell et al., 2018). These existing analyses indicate that the widespread declines in TWS

values and their hydrological components are primarily attributed to the intensive over-extraction of groundwater or warm-induced surface water loss, which are consistent with the findings in this study.

**4.2 Possible mechanisms of TC influence on TWS variability**

Periodic variability in the climate system can strongly influence regional meteorological patterns and their associated TWS. Unlike a single meteorological variable, teleconnection patterns control heat, moisture, and momentum balances through their effects on temperature, precipitation, and solar radiation reaching the Earth's surface (Zhu et al., 2017; Ni et al., 2018). Therefore, the inherent mechanisms of the TCs' influence on TWS variations are related to the combined simultaneous effects of TCs on regional climate factors (precipitation, temperature, and radiation); the changes in climate factors will

substantially affect the recharge (precipitation) and loss (evapotranspiration) of regional water resources, which eventually influence the changes in TWS. We have identified several dominant TCs that influence the variability in TWS and its components. Spatially, ENSO mainly controls the TWS variation over Southeast Asia, Southeast China, and India. During positive ENSO phases, warmer and drier conditions can easily occur over these regions. Higher temperatures and lower precipitation are both associated with an eventual decrease in TWS in these areas (Ni et al., 2017). IOD is similar to ENSO

and often co-occurs with ENSO (Du et al., 2009). During a positive IOD phase, anomalous cool (warm) waters appear in the eastern (western) Indian Ocean in association with large-scale circulation changes that bring anomalous dry conditions to Southeast Asia, i.e., Indonesia, while East Africa experiences above-average rainfall (Webster et al., 1999). The IOD may exert a negative impact on TWS due to the decrease of precipitation over Southeast Asia. Similarly, AO primarily dominates TWS variations in high-latitude areas and the surroundings of the Black Sea regions. When the AO index is positive, and the

vortex is intense, the winds tighten like a noose around the North Pole, locking cold air in place and contributing to unusual warmth over the Northern Hemisphere land masses (Zhou et al., 2001). This unusual warmth could lead to an increase in water loss through the evapotranspiration process, thereby contributing to a negative impact on TWS. The positive phase of the NAO, which is highly correlated with AO ($r = 0.64$, $p < 0.01$), leads to an intensification of the west wind drift due to a reinforcement of the Iceland low and the Azores high pressure systems; this phenomenon is particularly apparent in the

boreal winter months (Wallace et al., 1981). In turn, these results show positive precipitation anomalies in central Europe and negative anomalies in southern Europe and on the Norwegian coast, which is reflected in the water storage variations. Other TCs explain relatively small fractions of atmospheric variability in a given region by primarily interfering with hydrothermal processes. These results imply that climate variability may exert important influences on the TWS. Although previous studies mainly focused on the influence of ENSO on TWS (Phillips et al., 2012; Zhang et al., 2015; Ni et al., 2018),

a single indicator is unlikely to represent all climatic variability features over a large area (Zhu et al., 2017). In this study, we provide a comprehensive analysis of twelve commonly used teleconnection indices; these results indicate that the dominant TCs vary considerably for each component and region. Therefore, attributions and predictions of the changes of TWS and its components based on a single TC should be approached with caution, and multiple TCs are strongly recommended to explain the changes in TWS and its components.

## 4.3 Uncertainties and implications for future hydrological studies

Our results indicate that climate variability could explain the variability in TWS in most remote and sparsely populated regions. To a certain extent, climate variability may also indirectly explain glacial melt-induced changes in TWS, such as warming-induced glacial retreat. However, climate variability is influenced by human activities, such as groundwater abstraction in regions with intensive human activities. Although we obtained the contributions of different water storage components (SW, SM, and groundwater) through TWS partitioning, each component also influenced both the climate variability and human activities, which makes determinations of the influences of climate change and other processes extremely complicated. Thus, well-designed experiments and coupled human-natural system models are still needed to clarify the quantitative contributions of each influencing factor on TWS and its components' variability (Samaniego et al., 2010; Zhang et al., 2017; Rakovec et al., 2019). Several uncertainties also exist in understanding the changes in TWS and its components over the Asian and Eastern European regions. These may include the unaccounted reservoirs and rivers in surface water storage, which may induce uncertainties in a certain area in groundwater estimation by eliminating the surface water and soil moisture from TWS. The glacier data used in this study was obtained during 2000–2016, which was inconsistent with our study period (2002–2017); this incongruity may also have caused uncertainties in separating the water components from TWS. Additionally, the interdependencies among multiple teleconnections may further arise uncertainties in quantifying the relationship between TWS and TCs (Runge et al., 2019). Nevertheless, our study provides a new view of teleconnections that can enable a more thorough understanding of the changes in TWS. Moreover, our study focused primarily on a water storage deficit hotspot analysis because a basin-based evaluation may experience bias in calculating the basin-averaged TWS when a given basin simultaneously experiences drying and wetting trends in different sub-basins (Sun et al., 2018). A multiscale hydrological model with high spatial and temporal resolutions may help in understanding the effects of climate variability on the hydrological response across the globe. We infer that climate variability-induced extremes, such as drought and heatwaves, will exacerbate the TWS loss; this occurs through increased consumption of water resources from groundwater for irrigation and human water demand in these hotspots, rather than the climate variability alone being the sole cause for the observed TWS loss.

There are several recommendations for future hydrological studies: 1) withdrawal of freshwater from groundwater in water-limited regions is important for the sustainable development of water resources and food supplies (Rodell et al., 2018). However, groundwater drought is a distinct phenomenon resulting from a decrease in groundwater storage (Thomas et al., 2017). Understanding groundwater drought is important in water-limited regions where the interplay between groundwater recharge and abstraction results in variable groundwater stress conditions. GRACE has the unique potential to obtain data on groundwater storage by introducing subsidiary datasets. 2) Glacier mass loss in mountainous areas can relieve drought stress in drought years (Huss et al., 2018), but it can additionally result in hydroclimatic extremes, e.g., floods. Neither of these phenomena can be detected using only precipitation datasets, such as those commonly used in monitoring drought and flood events (Sherwood et al., 2014); this highlights the importance of TWS-related hydroclimatic extremes. With the release of

the GRACE follow-up satellite (Famiglietti et al., 2013), consecutive prolonged data records could provide a valuable solution for evaluating hydrological conditions from a long-term perspective and would lead to considerable improvements in our knowledge of TWS-related hydroclimatic extremes (Famiglietti et al., 2013). 3) A recent study found that the $CO_2$ growth rate is strongly sensitive to observed changes in TWS (Humphrey et al., 2018), and the coupling between the water and carbon cycles highlights the need for stronger interactions between the hydrological and biogeochemical research communities to achieve a sustainable development of the Earth system.

## 5 Conclusions

In this study, we characterize the spatiotemporal variations in TWS as well as its components and connect these variations with TCs over the Asian and Eastern European regions from April 2002 to June 2017 using multiple data sources. The results indicate a widespread decline in TWS during 2002–2017, and five hotspots of TWS negative trends were identified with trends ranging between -8.94 mm yr$^{-1}$ and -21.79 mm yr$^{-1}$. Partitioning of TWS suggests that these negative trends are mainly attributed to intensive groundwater extraction and warming-induced SW loss, but the contributions of each hydrological component vary from region to region. The results also indicate that ENSO, AO, and NAO are the three dominant factors in controlling variations in TWS through their covariability effects on climate variables. However, seasonal results suggest a divergent response of hydrological components to TCs among seasons and regions. This highlights the importance of knowledge of the seasonal responses to TCs to improve the understanding and prediction of changes in TWS and associated water storage components. Our study provides a comprehensive analysis of TWS variability and its connection to TCs across the Asian and Eastern European regions, thus facilitating the target strategy of water resource management.

## Data availability

The data and code generated in this study are available from the authors upon request (liuxianfeng7987@163.com).

## Author contributions

XL and XF conceived and designed the research, XL conducted the experiments and analysed the results, XL wrote the manuscript with contributions from XF, CP and BF.

## Competing interests

The authors declare that they have no conflict of interest.

**Acknowledgements**

The authors would like to thank the Goddard Earth Sciences Data and Information Services Center (http://grace.jpl.nasa.gov) and the Center for Space Research (http://www2.csr.utexas.edu/grace/RL05_ mascons.html) for providing global datasets of the JPL and CSR Mascon solutions and the total water content obtained from the Noah land hydrology model in the GLDAS v2 during 2002-2017. The authors would also like to thank the University of East Anglia Climatic Research Unit for providing the precipitation, temperature datasets, the European Centre for Medium-Range Weather Forecasts for providing evapotranspiration data, the Database for Hydrological Time Series of Inland Water for providing the sea level data, and the Climate Prediction Center of the U.S. National Oceanic and Atmospheric Administration for providing teleconnection indices. Comments of the editor and three anonymous reviewers were greatly appreciated.

**Financial support**

This work was sponsored by the National Natural Science Foundation (Nos. 41991230, 41722104 and 41801333), the National Key Research and Development Program of China (2017YFA0604700), the Chinese Academy of Sciences (QYZDY-SSW-DQC025), the China Postdoctoral Science Foundation (2019M650859 and 2019T120142), and the Fundamental Research Funds for the Central Universities (GK201901009, GK202003068).

**Review statement**

This paper was edited by Luis Samaniego and reviewed by three anonymous referees.

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

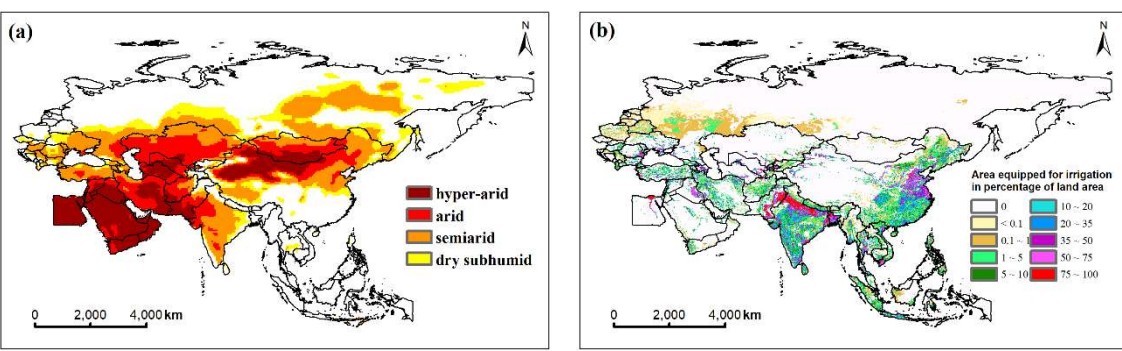

**Figure 1:** Boundaries of the Asian and Eastern European regions. Panel (a) is the spatial distribution of arid and semiarid areas based on the averaged aridity index during 2002–2017. The aridity index is calculated based on the ERA-Interim dataset downloaded from European Centre for Medium-Range Weather Forecasts. Panel (b) is the percentage area of irrigated land across the study area. The percentage area of irrigated land dataset is derived from Food and Agriculture Organization of the United Nations.

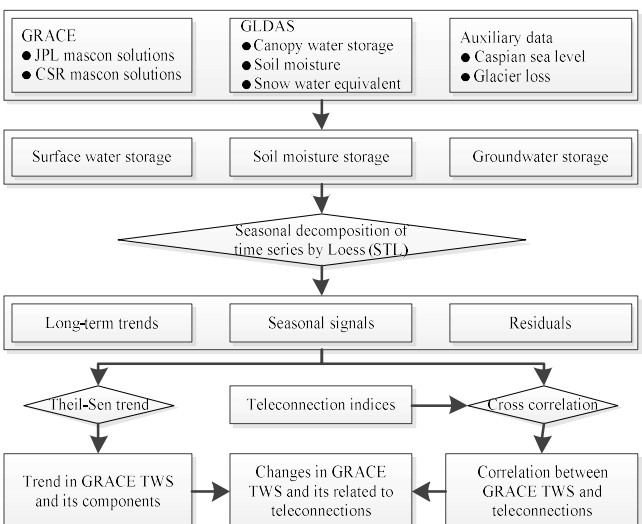

**Figure 2:** Methodology flow diagram of data processing in this study.

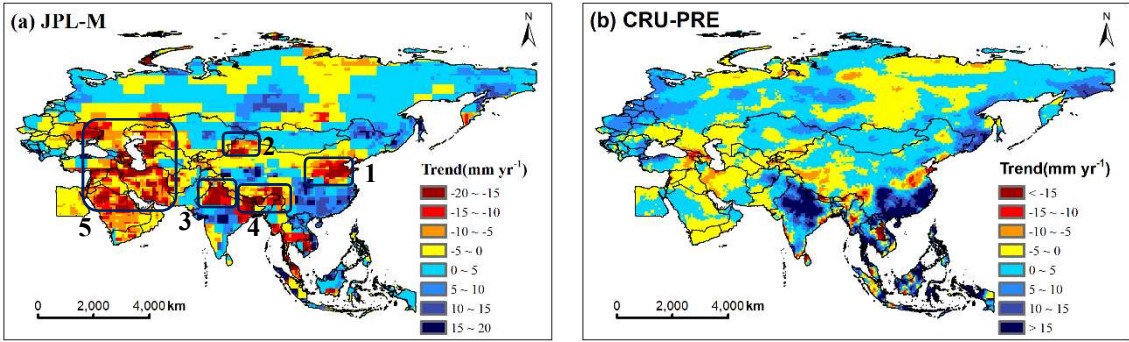

**Figure 3:** Spatiotemporal changes in TWS as obtained from GRACE (a) and precipitation as obtained from CRU (b) across the Asian and Eastern European regions during 2002–2017. The trend is obtained from the removed seasonal cycle time series.

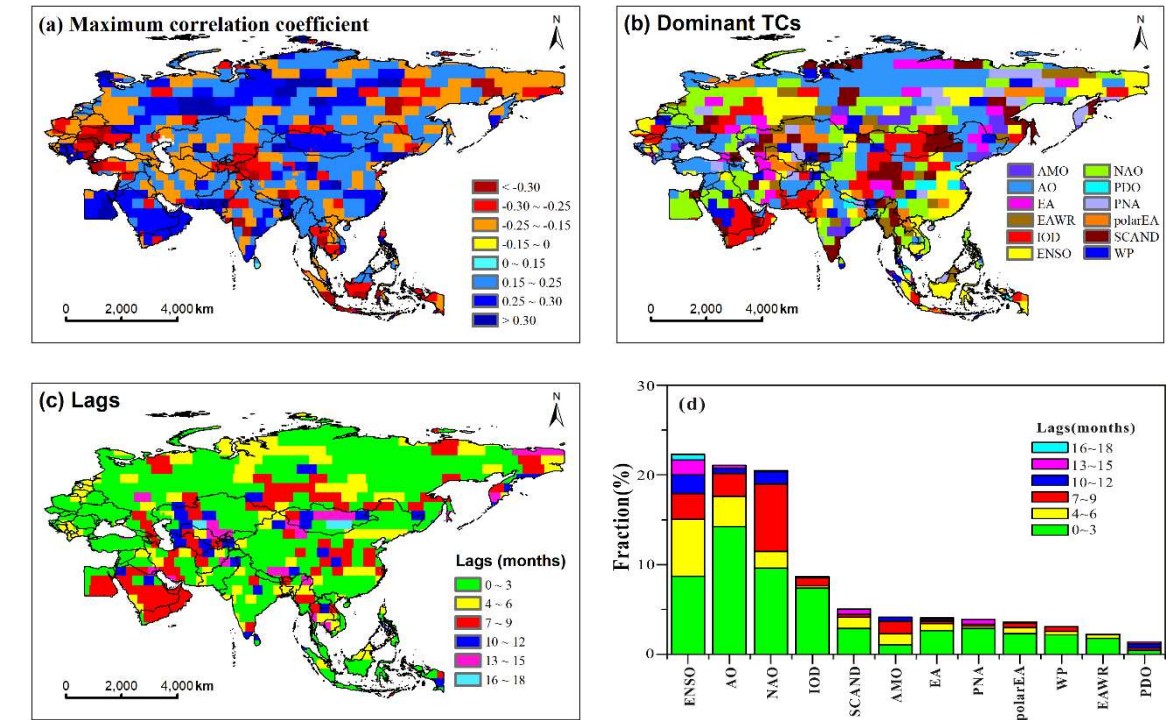

**Figure 4:** Spatial distribution of cross correlation analysis between TWS and teleconnection indices. (a) Spatial pattern of maximum correlation coefficients between TWS and the teleconnection indices. (b) Spatial pattern of teleconnections that can most accurately represent TWS variations. (c) Spatial pattern of teleconnection lag time. (d) Proportion of the area dominated by each teleconnection and its corresponding time lags. The maximum lag in the correlation analysis was limited to 0–24 months (significance threshold: |r| > ~0.15 given a significance level = 0.05 and a time series number = 183).

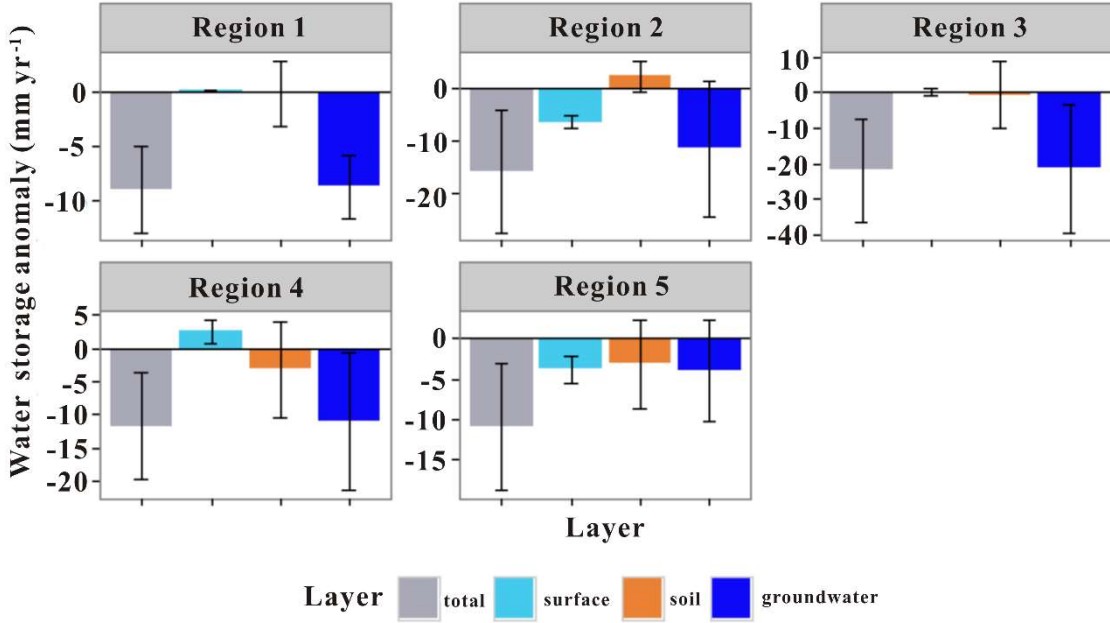

**Figure 5:** Contributions of different hydrological storages to TWS changes in five hotspots. Uncertainties represent 95% confidence intervals.

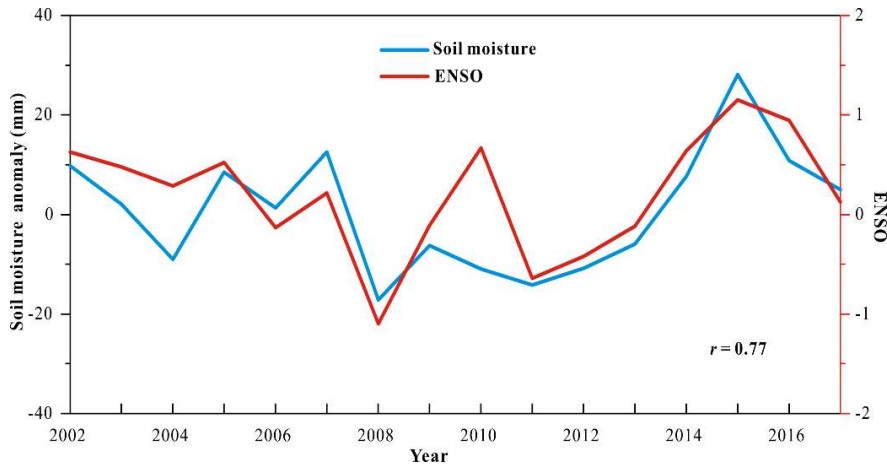

**Figure 6:** The residual time series of spring soil moisture and associated ENSO in region 3 during 2002–2017.

**Table 1:** Descriptions of datasets used in this study

| Datasets | Variables | Time span | Resolution | Source |
|---|---|---|---|---|
| GRACE | JPL-M | 2002-2017 | monthly and 0.5° | The Jet Propulsion Laboratory (Watkins et al., 2015) and the Center for Space Research (Save et al., 2016) |
| | CSR-M | | | |
| GLDAS | Canopy | 2002-2017 | monthly and 0.25° | The Global Land Data Assimilation System data (Rodell et al., 2004) |
| | Soil moisture | | | |
| | Snow water | | | |
| Lakes | Caspian sea | 2002-2017 | ten days and site | The Database for Hydrological Time Series of Inland Water (Schwatke et al., 2015) and Hydroweb (Crétaux et al., 2011) |
| | Aral Sea (East) | | | |
| | Aral Sea (West) | | | |
| | Aral Sea (North) | | | |
| Glacier | Tien Shan | 2000-2016 | year and regional | Literature (Brun et al., 2017) |
| | Hindu Kush | | | |
| | Spiti Lahaul | | | |
| | East Nepal | | | |
| | Bhutan | | | |
| | Nyainqentanglha | | | |
| Teleconnections | AO, NAO, EA, EAWR, WP, polarEA, PNA, IOD, AMO, PDO, ENSO, SCAND | 2002-2017 | monthly and global | The Climate Prediction Center of the U.S. National Oceanic and Atmospheric Administration |