# Peer review of "Widespread Decline in Terrestrial Water Storage and Its Link to Teleconnections across Asia and Eastern Europe"

_Hydrology and Earth System Sciences, 2019_

## Referee Comment (RC1) · Anonymous Referee #1 · 8 Oct 2019

The manuscript "Widespread decline in terrestrial water storage and its link to teleconnections across Asia and Eastern Europe" by Liu et al., submitted to HESS, analyses the terrestrial water storage (tws) for regions with declining tws based primarily on GRACE, hydrological modelling data and literature values, links it to a huge number of teleconnections and separate tws both in seasonality and compartments and link it as well to teleconnections. While the manuscript started promising (and the idea of linking TWS dynamics to teleconnections is interesting), it has several drawbacks both structural and content-wise. Simultaneously, I have the impression that the manuscript was not prepared carefully and properly reviewed by the co-authors before the submission. Otherwise I could not understand the number of the major and minor very obvious

problems that made it hard to focus on the content of the manuscript. In sum, I have doubts, if a major revision could lead to an acceptable improvement for the high journal standard and therefore recommend to reject the manuscript but I of course leave it up to the editor if the chance for improvement should be given.

Major comment: The general objective of the paper is interesting (especial the link to teleconnections) but how the authors structured the manuscript is not convincing. The method section does not provide the details that are needed to understand the results. Should the reader know every single teleconnection? What are the methodological details of assessing water storage changes for lakes (e.g. are reservoirs included?), how are glaciers included (a reference to literature does not allow a reader to really get a clue how specifically the data has been included in this study)? Wetland and river storage seem to be missing at all in the study – at least those are not indicated in the definition or in data sets used. The results section contains a too short and selective description of the results, often followed by an interpretation. Should it be up to the reader what is the result of the study or the interpretation? There are questionable interpretation included, for example that the (very small) changes in precipitation is responsible for the (one magnitude higher) change in tws, or that glacier melt leads to soil moisture increase – without citing any reference. In the discussion section, the arguments of the results section are partly repeated. The authors are not embedding the findings of their study to the literature (except a very few examples), so it is hard to get a proper information of the robustness of their findings. Most disappointing I found is that for nearly every figure, major problems arise. Most of the diagrams do not even have a proper axis naming / labelling, so I have hard times to understand the results and the text that is based on it, all that made it hard to review the content. More specific, there are (other than mentioned in the state of the art) already a number of global / large scale studies that deal with those or a subset of those regions or even on global scale but often directly include anthropogenic impacts (by the way, those regions could have names), e.g. Wada et al, 2010, Döll et al, 2014, Scanlon et al, 2018, 2019, Syed et al, 2008, Tangdamrongsub et al., 2018, Zhang et al, 2017 and more, those and some of

the references therein should be considered when re-designing the manuscript.

Specific comments: For the distinguishing of water storage compartments, a single mascon-solution and a single hydrological model is being used. Few years earlier that would have been state of the art, but now, and especially as a number of GRACE solutions (spherical harmonics and masons) and a large number of hydrological / land surface models are available, this kind of study should be done in a multi-model/multi-data setting to be able to verify the results, provide uncertainty information which then might lead to a valuable scientific contribution. To reduce the approach of the manuscript it to the minimum, the GRACE tws was reduced by NOAH soil moisture, snow and canopy, by lakes and glaciers; the leftovers are then groundwater and/or human interventions. Why have not the authors used a hydrological model (or better more) that consider human interventions, to allow direct assessment of trends / residuals? There are a number of global-scale studies that are using GRACE data in combination with global water models (Scanlon et al., 2018, Döll et al., 2014), especial to trends which contains also a huge list of references within for some of the regions of this study.

Line-by-line specific comments (not a full list, only the major things I stepped over during reading):

Line 72: The Mount Kilimanjaro comes unexpected in this list – isn't it located in Tansania (Africa), or is there also one in Asia?

Line 75: The sentence "Under the combined..." needs references or does it belong to the hypotheses?

Line 79ff: GRACE data, especially in the months at the end of the orbit time shows an increasing error in the signal – have you considered this in your analyses?

Lines 86-94 should be rewritten as it is repeating partly itself

Line 95: Whereas I agree that two things are comparable in general, please be concise in wording. One can compare an apple with an orange but this is not a good comparison. Comparing full TWS from GRACE with TWS from Noah that consists only of soil, snow and canopy leaves out important compartments such as water bodies, groundwater and glaciers. Of course, this is written in the next sentence but the word "directly comparable" is misleading.

Lines 98 ff: the description of how lake level and glacier change have been used in this study is much to short described. For lake levels – which lakes are included? Only the large ones? Are reservoirs included? Are wetlands included? Which time series are assessed? For example, Wang et al., (2018) ends in 2016, the time series of this manuscript exceeds this.

Line 101: If SW does not include wetlands or rivers (at least this information is missing in the manuscript), then the residual of GRACE TWS minus SW and SM cannot be groundwater only.

Lines 105 ff: The description of the TCs is not very informative. Please provide more details, e.g. for which region they are defined, how they are characterized (e.g. briefly in the supplement).

Lines 113 f: to which TWS does the section refers to? I guess to GRACE TWS, right? The section needs to be reformulated and streamlined for better readability and enriched by references, it reads confused in the current shape. What does the (total-trend-seasonality) mean? Is it a mathematical equation? Please provide details why by using the cross-correlation of the TWS residuals and TC the interference with (…) are reduced. This is similarly repeated in lines 144 f.

Line 144: For which GRACE solution the numbers are standing for? The mean of both? Fig 2c shows not "expected" changes in precipitation. And again, such a small precipitation trend in that region as shown in Fig 2b should not affect the tws signal drastically. Similar interpretation problems are following for the next case studies.

Line 158: The comparsion of Nort-West-India with one single reference is misplaced in

the results section. Due to the reason the authors explain, it is not possible to assess the reason for the difference. I suggest to properly frame the trends into the various estimates that are available from the literature and then, in the discussion section of the paper to discuss it.

Line 161: What is the assessment of Caspian Sea Level is based on? Is that focus of the paper?

Line 163 ff: A mix of (selected) interpretation and presenting results, not easy to follow.

Line 169 ff: It is hard to accept that general conclusion that change in tws correlates with natural variability just because of (the magnitude lower) precip trend. This needs to be analysed in much more detail, especially the role of human interventions needs to be considered here (with data/modelling).

Line 170 f: A data product that base on the same satellite input but with a different processing is expected to lead to similar results (at least for the broad picture) especially for the highly human impacted regions. This does not allow justification of the results in my eyes. It could provide an uncertainty information, not more. A different measurement system (e.g. GPS displacement analysis) could be a real justification.

Lines 182 f (Most regions. . .): I do not agree to the described pattern.

Lines 194 ff: it reads like a new finding that at those locations, groundwater depletion occurs. There is a wide range of previous literature that directly assess regions with groundwater depletion based on GRACE (and hydrological models), e.g. Döll et al., 2014, Wada et al., 2010 and references therein.

Line 199 f: is there any reference that the glacier melt leads to higher soil moisture or is it an interpretation of the results? I am not an expert in glacier hydrology but would assume that the effect of a melting glacier to soil moisture increase is only locally effective and as soon as the glacier water is within a river, soil moisture is affected probably only weak, especially at a larger spatial scales.

Line 202: irrigated agriculture contributes to more than a half of of tws loss? How has this been assessed? Is assumed that irrigation only stems from groundwater resources? The following lines are already a discussion, it is hard to assess what is the specific contribution of this study.

Line 208: the authors refer to a meteorological drought the first time in the manuscript. Is it referring to declining precipitation from Fig 2b? Trends in precipitation does not necessarily imply a drought, this should be clarified.

Line 210: again, everything is comparable. But not everything is similar/equal. Please be concise with wording.

Line 214: which drought definition? TWS is not "recharged", groundwater can be recharged. What does the word "will" mean? Climate projection? Water use projection? This is not clear.

Line 241: unit?

Section 3.2: I have hard times interpreting and justifying the results. First, maximum correlations are relatively low (Fig. S5) and I guess, only the TC with the dominant correlation is displayed in Fig 2. However, how to interpret plausible, if a correlation coefficient is, let's assume 0.20 and the next TC has 0.19? The interpretation (such as time lag discussion) solely considers the maximum correlation even though it is in a large part of the study area very low. A correlation coefficient of 0.2 implies that this specific TC explains 20% of the TWS signal, is this correct? This needs more attention and maybe cutting out dominant TCs below a meaningful threshold.

Section 4.1 repeats mainly the interpretation of the results section. The last paragraph does not provide any scientific insights in terms of a discussion.

Section 4.2 is a description of the TC and in last two sentences it is stated that those TCs are impacting TWS. The reader does not have a much better idea how TWS is affected. And yes, there are methodological questions to solve.

Line 297 f: what is meant with TWS dynamics attributions? I fully agree that coupled human-natural approaches have to be done to better understand to which part TWS dynamics are due to natural or due to anthropogenic variations. This could be then connected with a link to TCs.

The arrangement of Figures is not consistent. Fig 2f is referred to before 2c-e, Figure S6 is referred to before referring to S3 etc. Please follow the journal guidelines which improves the readability. It seems that Fig S6 is the same like Fig 2f – is there any reason for this repetition? Fig. 2e is not referred to in the manuscript.

Fig 1 and lines ∼75: sources are missing for definition of humidity and for area equipped for irrigation

Fig 2a and b and line 149 ff: I try to make sense out of the numbers and colours. TWS trend seems to be a magnitude larger then precipitation trend. How does a precipitation change of < 1 mm/yr can be the cause for 10 to 20 mm tws change? Precipitation can be a cause, yes, but if the numbers are correct, then I cannot agree that this is the reason and similarly I not agree that there where the pattern looks differently, human impact is the (only) reason. This needs by far more discussion and thorough analysis. From Table S1 some differences are visible for the two Mascon solutions. I suggest to display the two Mascon solutions in Fig 2. The regions in Table S1 could get names.

Fig 2c: check spelling of header text

Fig 2f: a legend is missing, and I can only see 4 lines and a mess of shaded area which does not allow any meaningful assessment. Please re-arrange (e.g. splitting it up to 5 single plots with same Y-axis) and it would be meaningful to use month/years for x-axis.

Fig 3: Labelling of Y-Axis with "Water loss" and then negative values – does it imply a water gain? Please name it more meaningful.

Fig 4: what can be seen at both axis? It seems that the months are not consecutive (If I

[Figure]

interpret it correctly as spring season), then drawing a solid line through it is misleading.

Fig S1: unit for Y-Axis is missing. I suggest to use month/years instead of month numbers. Why does the time series ends ∼ at month 165 whereas the other figures are ending at month ∼177/181?

Fig S3: what is shown at X- and Y-Axis? I have not checked if the references are listed in the reference list and vice versa, and also have not checked the reference list itself.

References Döll, P., Müller Schmied, H., Schuh, C., Portmann, F. T., & Eicker, A. (2014). Global-scale assessment of groundwater depletion and related groundwater abstractions: Combining hydrological modeling with information from well observations and GRACE satellites. Water Resources Research, 50(7), 5698–5720. https://doi.org/10.1002/2014WR015595

Scanlon, B. R., Zhang, Z., Save, H., Sun, A. Y., Müller Schmied, H., van Beek, L. P. H., et al. (2018). Global models underestimate large decadal declining and rising water storage trends relative to GRACE satellite data. Proceedings of the National Academy of Sciences, 201704665. https://doi.org/10.1073/pnas.1704665115ˆ

Scanlon, B. R., Zhang, Z., Rateb, A., Sun, A., Wiese, D., Save, H., Beaudoing, H., Lo, M. H., Müller Schmied, H., Döll, P., van Beek, R. Swenson, S., Lawrence, D., Croteau, M., Reedy, R. C. (2019). Tracking seasonal fluctuations in land water storage using global models and GRACE satellites. Geophysical Research Letters 46 (10), 5254-5264, 10.1029/2018GL081836

Syed T.H., Famiglietti J.S., Rodell M., Chen J., Wilson C.R. (2008). Analysis of terrestrial water storage changes from GRACE and GLDAS. Water Resour Res 44:W02433

Tangdamrongsub, N., Han, S.-C., Tian, S., Schmied, H. M., Sutanudjaja, E. H., Ran, J., & Feng, W. (2018). Evaluation of groundwater storage variations estimated from GRACE data assimilation and state-of-the-art land surface models in Australia and the North China Plain. Remote Sensing, 10(3). https://doi.org/10.3390/rs10030483

Wang, J., Song, C., Reager, J.T., Yao, F., Famiglietti, J.S., Sheng, Y., MacDonald, G.M., Brun, F., Müller Schmied, H., Marston, R.A., Wada, Y. (2018). Recent global decline in endorheic basin water storages. Nature Geoscience 11, 926-932, doi: 10.1038/s41561-018-0265-7.

Zhang L, et al. (2017) Validation of terrestrial water storage variations as simulated by different global numerical models with GRACE satellite observations. Hydrol Earth Syst Sci 21:821–837.

---

## Referee Comment (RC2) · Anonymous Referee #2 · 14 Oct 2019

The manuscript titled "Widespread decline in terrestrial water storage and its link to teleconnections across Asia and Eastern Europe" by Liu et al., has identified an interesting research gap of analyzing the linkage between teleconnections (TCs) with terrestrial water storage (TWS) in Asia and Eastern Europe. They have utilized comprehensive set of TCs for the study. The TWS has been abstracted from GRACE observations. The TWS is partitioned using GLDAS to generate surface water (SW), soil moisture (SM) and groundwater. The TWS components are then de-seasonalized. This is followed by spatio-temporal trend analysis, comparison analysis with TCs and dissection of each TWS component's contribution to TWS.

Although the manuscript embeds a promising research topic, the level of write up lags far behind the study done which in turn lags behind the research gap stated. The manuscript lacks crisp, clear messages. Most of the time this is due to poor sentence structure and grammar. The reader has to infer what the authors are trying to state or sometimes even conclude. I would not recommend to accept the manuscript in its current form and structure. I would suggest the following major revisions to the authors, if the editor decides to move the process forward -

Major Comments -

- It is mentioned that the lead author wrote the manuscript with contributions from all others. However, there are significant improvements required in the sentences, paragraphs and information sequence structure. This shows the manuscript was not sufficiently revised before submission. In its current form, the manuscript doesn't facilitate an uninterrupted flow.

- Tabulation of data information and a flow diagram of data processing are necessary in the section "Data". It might be a better idea to make an overall methodology flow diagram that includes this.

- There is no background on TCs and subject matter of the manuscript in regards to the study area in the section "study area".

- Although the subtopic headings of results section are clear, the content is very poorly structured and sequenced. Smooth reading flow is missing.

- Although the subtopic headings of discussion section are clear, the content is very poorly structured and sequenced. For an instance, in section "4.2 Possible mechanisms of TC influence on TWS variability", only the dominant TC and their role in the regions are mentioned. However, the result from this study and discussion on "possible linkage" with these "roles" are missing.

- The figures and tables of the supplementary section are directly and heavily referred

to in the manuscript. This needs to be ratified. The grouping of figures is also not optimal.

- The preparation of figures and tables is poor and needs significant revising. This includes axes labeling, color combination, color code, headers, data source declaration in the captions.

Specific Comments (in order of sections of the draft)-

1 Introduction

Line 38: May be the gap is mainly in Asia. There are existing studies for Europe. E.g. Rakovec et al. (2016) have already analyzed the TWS anomaly using GRACE in 400 European river basins.

Line 42: "...are undergoing intensive..."

Line 44: mm/y vs mm yr-1 (in abstract). Inconsistent usage of unit format.

Line 63-64: "...and the remainder of the TWS time series...". Its not clear for the reader what this is referring to.

2.1 Study Area

Figure 1a: Source/ citation is missing. Color representation for Semi-arid and dry sub-humid regions don't have sufficient contrast.

Line 72: Mount Kilimanjaro is in Africa, not the study area.

Line 73: The URL link doesn't have public access. And its mentioning here is also not clear.

Line 75: I couldn't grasp the usage of "...in this area". Did you mean to say "More importantly, the Asian and Eastern European regions are the most densely populated regions in the world, sustaining nearly half of the global population and contain some of the largest and most intensively irrigated lands of the world"? If so, kindly cite the

source.

Figure 1b: Source is missing.

Line 75-77: Split long sentence to two. Also kindly include the source.

Line 69-77: The research gap of comprehensive TWS-TC correlation would need some background on TCs in the study area. This is missing and should be discussed in more detail in this section. If the manuscript space permits, additional map/s depicting the TCs and their role in the regions would improve the clarity of the topic to the readers.

2.2 Data

Line 79-110: Tabulating the dataset information would be a more efficient way of presenting than the current form. E.g. Line 91 mentions the exact same thing mentioned in previous sentence just adding additional spatial information. This is followed by the web URL for the dataset. Tabulation of info would be the perfect solution here.

Line 92: Avoid using direct URLs for data reference. There are better ways to cite datasets.

Line 93: Bicubic interpolation doesn't preserve the mass while resampling. Mass-conservative remapping is advised (e.g. remapcon operator of CDO)

Line 94-97: This sentence has 62 words! This doesn't help readability of the manuscript. Kindly break into shorter sentences. One message per sentence.

Line 98: The x-axis header for Figure S1 is missing. Moreover, the x-axis is not uniform across figures S1, S2, S3, S6.

Line 101: "... by deducting..."

Line 79-110: Apart from tabulating the dataset information, the data assimilation approach would be much easier presented as flow diagram.

Line 105-110: This can go into the data table. However, some elaboration of these 12
TCs in regards to the study area is missing. I would suggest to switch the position of 2.1 and 2.2. With this new sequence, the authors can provide further insight on TCs from view point of study area in the section "Study Area".

2.3 Methods

2.3.1 Time series decomposition

Line 113: The first sentence here is out of subject. The current subject is decomposition and this sentence is about trend analysis (belongs to section 2.3.2?)

Line 113-124: Revise the order of the sentences and info. Not in optimum order.

Line 118: STL is a robust method? Then cite the papers who have proven this method to be robust.

Line 118: "for detecting non-linear time series in trend estimates". What do you mean? The sentence doesn't make sense to me.

Line 123-124: Refer to previous publication of this journal to understand how to cite in such situation.

2.3.2 Theil-Sen trend analysis

Line 126: "..the linear trend of .... and precipitation for the ..."

Line 126: Trend analysis on the deseasonalized time series? or the residuals? I am guessing the first sentence of section 2.3.1 belongs here [??]

Line 126-127: Break the long sentence into two. Move the second part to the end of the section. In this way flow of read regarding Theil-Sen trend analysis is maintained.

Line 128: "... of the Theil-Sen trend analysis is ..."

Line 130: Remove "non-robust". And cite the literature proving this statement.

Line 130: "The TWS trend, ß, for a ..."

**2.3.3 Cross-correlation analysis**

Line 136: TWS or TWS residual?

Line 139: Move the "," to the end of the equation. At the moment its on the denominator.

Line 139: What is the meaning of the symbol Tau here? Is it the lag?

Line 140-141: "auto-covariace"?

Line 142: Why between 0 and 24? Reason, in brief, required. Moreover, can cross-correlation have value greater than 1?

Line 143-146: Break the sentence to get one message per sentence.

**3 Results**

**3.1 Spatiotemporal changes in TWS**

Line 149: Mention that both JPL-M and CSR-M showed similar spatio-temporal pattern to begin with. Then let the reader know that the values will be referred to JPL-M.

Line 152: The 5 hotspots are clearly shown in Figure 2a. This should be included in the reference illustrations.

Line 152: Figure 2f doesn't have color code. Unit of time is missing. Time axis with years as axis tick labels would enhance clarity.

Line 153: Figure S6 doesn't have color code.

Line 154: Precipitation is in figure 2b, not 2c.

Line 157: Precipitation is in figure 2b, not 2c.

Line 158: How is this "within" Rodell's findings? The estimates are completely contrasting each other. Plus, the reasoning is not convincing and/or explained properly.

Line 162: -73.2 mm y-1 is out of range for figure 2a.

3.2 Influence of TC indices on TWS variability

Line 174: Which section of the multi-plot Figure 2? Revise the header and legend header of figure S5 to something more explicit. In caption of Figure S5, replace "phase shift" by "lag". Usage of same terminology maintains the flow of reading. And what are the symbol alpha and "n" in the caption?

Line 175: "... and NAO have a significant area of influence on TWS variability."

Line 181: Resolve the structural error. Two sentences or one?

Line 189: "Proportions of time .... Figure 2d". This sentence should be the starting sentence of this paragraph.

Line 190: Tibetan plateau and Mongolia have more pixels of longer lags than SE Asia.

3.3 Contributions of water storage components to TWS

Line 194: The first sentence is concluding findings. Thus, it is more suitable to be placed towards the end of the section.

Line 195: The hotspots have been well established in the manuscript and doesn't need the reference to Figure 2a every time.

Line 195: "Groundwater depletion dominates the contribution to TWS loss in region 1..."

Line 197: "Similar results were observed in northwest..."

Line 210: The term "sea level" could be confused with the world mean sea level. Suggested rephrasing: "...Caspian Sea with a decrease in its water level elevation by -73.2 mm y-1..."

3.4 Divergent response of water storage components to TCs

Line 217: The first sentence is concluding findings. Thus, it is more suitable to be placed towards the end of the section.

Line 222: "...is a synthesis signal i.e. its trend ...". "... different ways. Furthermore, the groundwater ..."

Line 223: "...which indicates lower correlation..."

Line 225: Reference to a figure or table missing

4 Discussion

4.1 Comparison of our results to previous studies

Line 235-257: Please be clear about which region are you talking about. If you start with "Region 1 shows ..." then its clear that the information corresponding to that region (1, 3 and 4) else its hard to follow (regions 2 and 4).

Line 258-260: Break the sentence to get one message per sentence.

Line 260-263: Break the sentence to get one message per sentence.

Line 263-264: Revise the sentence for clarity. The concluding sentence should be clear.

4.2 Possible mechanisms of TC influence on TWS variability

Section 4.2: This is probably the most interesting subtopic of the paper. The writing style should have followed this pattern: 1) result observation at each hotspot, 2) literature on the dominant TC for the hotspot, 3) linking "possible mechanism" between the literature and the results. Currently the section is filled with only 2. There is no linking going on.

4.3 Implications for future hydrological studies

Line 288: "... could explain the variability in TWS in most of the remote and ..."

Line 290: "... variability interacts with human..."

Line 297: "claim" is a very strong word. Moreover, it doesn't make sense especially

as the paper hasn't included any prediction or scenario analysis of droughts and heat-waves.

Line 300-314: Usage of "First, Second, Third" in paragraph structure requires different approach of writing. Instead, bullet style enumeration of the three recommendations would suit the current sequence and structure of writing i.e. start first bullet with "Withdrawal of ...", and so on.

5 Conclusions

Line 321: "...component vary from region to region. The ..."

Line 323: "...and regions. This highlights the importance ...."

References:

RAKOVEC, O., KUMAR, R., MAI, J., CUNTZ, M., THOBER, S., ZINK, M., ATTINGER, S., SCHÄFER, D., SCHRÖN, M. and SAMANIEGO, L.: Multiscale and Multivariate Evaluation of Water Fluxes and States over European River Basins, J. Hydrometeorol., 17, 287–307, doi:10.1175/JHM-D-15-0054.1, 2016.

---

## Author Comment (AC1) · 1 Jan 2020

We would like to thank the reviewers for their professional, detailed and constructive comments, which improved our manuscript considerably. We have carefully revised the manuscript following their comments point by point. Our revisions and explanations have been inserted in blue, and all amendments are also highlighted in the version of revised manuscript. Additionally, the writing of our revised manuscript are also under carefully editing by English native speaker with specialized in hydrology. Anonymous Referee #1 The manuscript "Widespread decline in terrestrial water storage and its link to teleconnections across Asia and Eastern Europe" by Liu et al., submitted to HESS,

analyses the terrestrial water storage (tws) for regions with declining tws based primarily on GRACE, hydrological modelling data and literature values, links it to a huge number of teleconnections and separate tws both in seasonality and compartments and link it as well to teleconnections. While the manuscript started promising (and the idea of linking TWS dynamics to teleconnections is interesting), it has several drawbacks both structural and content-wise. Simultaneously, I have the impression that the manuscript was not prepared carefully and properly reviewed by the co-authors before the submission. Otherwise I could not understand the number of the major and minor very obvious problems that made it hard to focus on the content of the manuscript. In sum, I have doubts, if a major revision could lead to an acceptable improvement for the high journal standard and therefore recommend to reject the manuscript but I of course leave it up to the editor if the chance for improvement should be given. Response: Thank you for your comment. We feel sorry for the confusion and inconvenience we have brought to you. In the revised manuscript, we have substantially revised our manuscript according to the reviewers' comments. Major comment (1) The general objective of the paper is interesting (especial the link to teleconnections) but how the authors structured the manuscript is not convincing. Response: Thank you for your comment. We have reorganized the data and method, result and discussion section according to referee's comments, particularly in the result interpretation and discussion content. (2) The method section does not provide the details that are needed to understand the results. Should the reader know every single teleconnection? What are the methodological details of assessing water storage changes for lakes (e.g. are reservoirs included?), how are glaciers included (a reference to literature does not allow a reader to really get a clue how specifically the data has been included in this study)? Wetland and river storage seem to be missing at all in the study – at least those are not indicated in the definition or in data sets used. Response: Thank you for your comments. In our revised manuscript, we have tabulated the datasets used in our study. The lakes and glaciers that considered in our study are listed in the table (Table 1, see attached supplement, hereafter). The rivers and reservoirs indeed not included

in our study, we have discussed the associated uncertainties in discussion section. We also made a methodology flow diagram of data processing in our revised manuscript (Figure 2, marked by the figure number in the revised manuscript and attached supplement, hereafter). (3) The results section contains a too short and selective description of the results, often followed by an interpretation. Should it be up to the reader what the result of the study or the interpretation is? There are questionable interpretation included, for example that the (very small) changes in precipitation is responsible for the (one magnitude higher) change in TWS, or that glacier melt leads to soil moisture increase – without citing any reference. Response: Thank you for your comment. We have substantially modified the inappropriate phrasing in results interpretation, and also added citations for each interpretation. Notably, the trend in precipitation was mistake in our former version of manuscript, we have recalculated and reproduced the spatiotemporal changes of precipitation over the study area (Figure 3). (4) In the discussion section, the arguments of the results section are partly repeated. The authors are not embedding the findings of their study to the literature (except a very few examples), so it is hard to get a proper information of the robustness of their findings. Response: Thank you for your comment. We have reorganized the discussion section according to reviewers' comments in our revised manuscript. (5) Most disappointing I found is that for nearly every figure, major problems arise. Most of the diagrams do not even have a proper axis naming / labelling, so I have hard times to understand the results and the text that is based on it, all that made it hard to review the content. Response: Thank you for your comment. We feel sorry for the inconvenience we have caused to you. In the revised manuscript, we have reproduced all figures according to the detail comments. We have attached all figures at the end of this response. (6) More specific, there are (other than mentioned in the state of the art) already a number of global / large scale studies that deal with those or a subset of those regions or even on global scale but often directly include anthropogenic impacts (by the way, those regions could have names), e.g. Wada et al, 2010, Döll et al, 2014, Scanlon et al, 2018, 2019, Syed et al, 2008, Tangdamrongsub et al., 2018, Zhang et al, 2017 and more,

those and some of the references therein should be considered when re-designing the manuscript. Response: Thank you for your comment. We have carefully read these papers and properly cited them in our revised the manuscript. Specific comments (1) For the distinguishing of water storage compartments, a single mascon-solution and a single hydrological model is being used. Few years earlier that would have been state of the art, but now, and especially as a number of GRACE solutions (spherical harmonics and masons) and a large number of hydrological / land surface models are available, this kind of study should be done in a multi-model/multi-data setting to be able to verify the results, provide uncertainty information which then might lead to a valuable scientific contribution. To reduce the approach of the manuscript it to the minimum, the GRACE tws was reduced by NOAH soil moisture, snow and canopy, by lakes and glaciers; the leftovers are then groundwater and/or human interventions. Why have not the authors used a hydrological model (or better more) that consider human interventions, to allow direct assessment of trends / residuals? There are a number of global-scale studies that are using GRACE data in combination with global water models (Scanlon et al., 2018, Döll et al., 2014), especial to trends which contains also a huge list of references within for some of the regions of this study. Response: Thank you for your constructive comment. The spherical harmonic solutions generally suffer from correlated errors that manifest longitudinal striping in the gravity solution (Rodell et al., 2018). Although largely successful in removing errors, the post-processing also damps and smooths real geophysical signals (Landerer and Swenson, 2012). Recent advances in GRACE data processing have shown that solving for gravity anomalies in terms of mass concentration (mascon) functions with carefully selected regularization results in superior localization of signals on an elliptical Earth (Save et al., 2016). Therefore, two publicly available GRACE mascon solutions are employed in our study: Jet Propulsion Laboratory mascons RL05M (Watkins et al., 2015) (JPL-M) and Center for Space Research mascons RL05M (Save et al., 2016) (CSR-M). Notably, JPL-M has the unique characteristic that each $3°$ mascon element is relatively uncorrelated with neighboring mascon elements, whereas the $1°$ mascon elements in CSR-M solutions

is highly correlated with their neighbors. Moreover, three degrees correspond approximately to the 'native' resolution of GRACE. Therefore, in this work we mainly used JPL-M for trend analysis and mapping. (2) Line 72: The Mount Kilimanjaro comes unexpected in this list – isn't it located in Tansania (Africa), or is there also one in Asia? Response: Thank you for your comment. The Mount Kilimanjaro is indeed located in Africa, we have corrected the mistake in our revised manuscript. (3) Line 75: The sentence "Under the combined: : :" needs references or does it belong to the hypotheses? Response: Thank you for your comment. We have rewritten the study area section and deleted this sentence in our revised manuscript. (4) Line 79ff: GRACE data, especially in the months at the end of the orbit time shows an increasing error in the signal – have you considered this in your analyses? Response: Thank you for your comment. There are indeed certain months during which the GRACE orbit is in a near-repeat pattern. This phenomenon leads to sub-optimal spatial sampling and thus typically leads to larger errors in the higher spherical harmonic coefficients. The mascon solutions used in this study have already considered the measurement errors and leakage errors in the final data analyses data product. (5) Lines 86-94 should be rewritten as it is repeating partly itself Response: Thank you for your comment. We have rewritten the data section in our revised manuscript. (6) Line 95: Whereas I agree that two things are comparable in general, please be concise in wording. One can compare an apple with an orange but this is not a good comparison. Comparing full TWS from GRACE with TWS from Noah that consists only of soil, snow and canopy leaves out important compartments such as water bodies, groundwater and glaciers. Of course, this is written in the next sentence but the word "directly comparable" is misleading. Response: Thank you for your comment. We have rewritten this section and revised the word "directly comparable" in our revised manuscript. (7) Lines 98 ff: the description of how lake level and glacier change have been used in this study is much to short described. For lake levels – which lakes are included? Only the large ones? Are reservoirs included? Are wetlands included? Which time series are assessed? For example, Wang et al., (2018) ends in 2016, the time series of this manuscript exceeds this. Response: Thank you

for your comment. We have listed the lakes and glaciers used in our study in table 1. But we did not include reservoirs and rivers parts in our study. We have discussed the associated uncertainties in discussion section as follows. Multiple uncertainties remain in understanding the changes in TWS and its components over the Asian and Eastern European regions. These may include the unaccounted for reservoir and rivers in surface water storage, which may induce uncertainties in a certain area in estimating the groundwater by deducting the surface water and soil moisture from TWS. The glacier data used here is during 2000-2016, this inconsistent with our study period (2002-2017) may also cause uncertainties in separating the water components from TWS. (8) Line 101: If SW does not include wetlands or rivers (at least this information is missing in the manuscript), then the residual of GRACE TWS minus SW and SM cannot be groundwater only. Response: Thank you for your comment. We indeed not consider rivers and reservoirs parts in our study. We have added the uncertainties in discussion section in our revised manuscript. (9) Lines 105 ff: The description of the TCs is not very informative. Please provide more details, e.g. for which region they are defined, how they are characterized (e.g. briefly in the supplement). Response: Thank you for your comment. We have supplemented the briefly introduction of the TCs in data section in our revised supplement. (10) Lines 113 f: to which TWS does the section refers to? I guess to GRACE TWS, right? The section needs to be reformulated and streamlined for better readability and enriched by references, it reads confused in the current shape. What does the (totaltrend-seasonality) mean? Is it a mathematical equation? Please provide details why by using the cross-correlation of the TWS residuals and TC the interference with (: : :) are reduced. This is similarly repeated in lines 144 f. Response: Thank you for your comment. Yes, this section refers to GRACE TWS, we have revised the statement. Also, we have reformulated and streamlined this section according to your useful comment in our revised manuscript. (11) Line 144: For which GRACE solution the numbers are standing for? The mean of both? Fig 2c shows not "expected" changes in precipitation. And again, such a small precipitation trend in that region as shown in Fig 2b should not affect the tws signal drastically. Sim-

ilar interpretation problems are following for the next case studies. Response: Thank you for your comment. Both JPL-M and CSR-M show similar spatiotemporal pattern of changes in TWS (Figure 3 and Figure S3). Since the JPL-M solution has the merit of lack of correlation between neighboring mascon elements in the retrieval, in this work we use JPL-M for trend analysis and mapping. Notably, the trend in precipitation was mistake in our former version of manuscript, we have recalculated and reproduced the spatiotemporal changes of precipitation over the study area. (12) Line 158: The comparsion of Nort-West-India with one single reference is misplaced in the results section. Due to the reason the authors explain, it is not possible to assess the reason for the difference. I suggest to properly frame the trends into the various estimates that are available from the literature and then, in the discussion section of the paper to discuss it. Response: Thank you for your comment. We agree with your suggestions, and we have revised the sentences according to the comment in our revised manuscript. (13) Line 161: What is the assessment of Caspian Sea Level is based on? Is that focus of the paper? Response: Thank you for your comment. In this paper, we estimated the surface water loss by assessing the decline in water body level of Caspian Sea. The sharply declined in Caspian Sean level could better understand the loss of surface water storage. (14) Line 163 ff: A mix of (selected) interpretation and presenting results, not easy to follow. Response: Thank you for your comment. We have redesigned this paragraph in our revised manuscript. (15) Line 169 ff: It is hard to accept that general conclusion that change in tws correlates with natural variability just because of (the magnitude lower) precipitation trend. This needs to be analysed in much more detail, especially the role of human interventions needs to be considered here (with data/modelling). Response: Thank you for your comment. The trend in precipitation was mistake in our former version of manuscript, we have recalculated and reproduced the spatiotemporal changes of precipitation over the study area. Challenges remain in separating the long-term relative roles of natural climatic variation and anthropogenic forcing on TWS changes. Well-designed experiments and coupled human-natural system models are still needed to clarify the quantitative contributions of each influencing

factor on TWS in our future study. (16) Line 170 f: A data product that base on the same satellite input but with a different processing is expected to lead to similar results (at least for the broad picture) especially for the highly human impacted regions. This does not allow justification of the results in my eyes. It could provide an uncertainty information, not more. A different measurement system (e.g. GPS displacement analysis) could be a real justification. Response: Thank you for your comment. We have rephrased this sentence, and rewritten the results section in our revised manuscript. (17) Lines 182 f (Most regions: : :): I do not agree to the described pattern. Response: Thank you for your comment. We have revised this statement in our revised manuscript. (18) Lines 194 ff: it reads like a new finding that at those locations, groundwater depletion occurs. There is a wide range of previous literature that directly assess regions with groundwater depletion based on GRACE (and hydrological models), e.g. Döll et al., 2014, Wada et al., 2010 and references therein. Response: Thank you for your comment. We have carefully read these papers and properly cited in our revised manuscript. (19) Line 199 f: is there any reference that the glacier melt leads to higher soil moisture or is it an interpretation of the results? I am not an expert in glacier hydrology but would assume that the effect of a melting glacier to soil moisture increase is only locally effective and as soon as the glacier water is within a river, soil moisture is affected probably only weak, especially at a larger spatial scales. Response: Thank you for your comment. In addition to the glacier melt water, the increase in precipitation could also contribute to the increase in soil moisture (Figure 3). We have revised this sentence in our revised manuscript. (20) Line 202: irrigated agriculture contributes to more than a half of tws loss? How has this been assessed? Is assumed that irrigation only stems from groundwater resources? The following lines are already a discussion, it is hard to assess what is the specific contribution of this study. Response: Thank you for your comment. Actually, groundwater contributes to more than a half of TWS loss in region2 instead of irrigated agriculture. We have rewritten this part in our revised manuscript. (21) Line 208: the authors refer to a meteorological drought the first time in the manuscript. Is it referring to declining precipitation from Fig 2b? Trends in precipitation does not necessarily imply a drought, this should be clarified. Response: Thank you for your comment. We indeed inferred drought from declining precipitation, and we have rectified the statement in our revised manuscript. (22) Line 210: again, everything is comparable. But not everything is similar/equal. Please be concise with wording. Response: Thank you for your comment. We have replaced the word "comparable" of "similar" in our revised manuscript. (23) Line 214: which drought definition? TWS is not "recharged", groundwater can be recharged. What does the word "will" mean? Climate projection? Water use projection? This is not clear. Response: Thank you for your comment. We have replaced the word "recharged" of "changed", and we also rephrased this sentence in our revised manuscript. (24) Line 241: unit? Response: Thank you for your comment. We have rectified the unit in our revised manuscript. (25) Section 3.2: I have hard times interpreting and justifying the results. First, maximum correlations are relatively low (Fig. S5) and I guess, only the TC with the dominant correlation is displayed in Fig 2. However, how to interpret plausible, if a correlation coefficient is, let's assume 0.20 and the next TC has 0.19? The interpretation (such as time lag discussion) solely considers the maximum correlation even though it is in a large part of the study area very low. A correlation coefficient of 0.2 implies that this specific TC explains 20% of the TWS signal, is this correct? This needs more attention and maybe cutting out dominant TCs below a meaningful threshold. Response: Thank you for your thoughtful comment. We indeed adopt the maximum correlation coefficient as the dominant TC. We also agree with your comment, and the situation mentioned above could occur in data processing. However, the pixel is independent each other. For each pixel, we could extract the maximum correlation coefficient between TWS and TCs, but we could not obtain the area proportion of each dominant TC during extraction process. Therefore, we adopted maximum correlations to interpretation, and we also discussed this uncertainty in discussion section of our revised manuscript. (26) Section 4.1 repeats mainly the interpretation of the results section. The last paragraph does not provide any scientific insights in terms of a discussion. Response: Thank you for your comment. We have reorganized the discussion section according to the both

reviewers' comments in our revised manuscript. (27) Section 4.2 is a description of the TC and in last two sentences it is stated that those TCs are impacting TWS. The reader does not have a much better idea how TWS is affected. And yes, there are methodological questions to solve. Response: Thank you for your comment. We have added the possible impacts of TCs on TWS according to reviewers' comments in our revised manuscript. (28) Line 297 f: what is meant with TWS dynamics attributions? I fully agree that coupled human-natural approaches have to be done to better understand to which part TWS dynamics are due to natural or due to anthropogenic variations. This could be then connected with a link to TCs. Response: Thank you for your comment. We have revised the statement in our revised manuscript. The coupled human-natural model is a promising and challenging issue that need pay more attention in our future work. (29) The arrangement of Figures is not consistent. Fig 2f is referred to before 2c-e, Figure S6 is referred to before referring to S3 etc. Please follow the journal guidelines which improves the readability. It seems that Fig S6 is the same like Fig 2f – is there any reason for this repetition? Fig. 2e is not referred to in the manuscript. Response: Thank you for your comment. We have reproduced all figures, and rearranged the sequence of figures in our revised manuscript. We have attached all figures at the end of this response. (30) Fig 1 and lines âĹij75: sources are missing for definition of humidity and for area equipped for irrigation Response: Thank you for your comment. We have supplemented the sources for definition of humidity and for area equipped for irrigation in figure caption. (31) Fig 2a and b and line 149 ff: I try to make sense out of the numbers and colours. TWS trend seems to be a magnitude larger then precipitation trend. How does a precipitation change of < 1 mm/yr can be the cause for 10 to 20 mm tws change? Precipitation can be a cause, yes, but if the numbers are correct, then I cannot agree that this is the reason and similarly I not agree that there where the pattern looks differently, human impact is the (only) reason. This needs by far more discussion and thorough analysis. From Table S1 some differences are visible for the two Mascon solutions. I suggest to display the two Mascon solutions in Fig 2. The regions in Table S1 could get names. Response: Thank you for your comment. We feel sorry

for the mistake in trend analysis of precipitation in our former version of manuscript, we have recalculated and reproduced the spatiotemporal changes of precipitation over the study area (Figure 3). (32) Fig 2c: check spelling of header text Response: Thank you for your careful comment. We have revised the spelling of header text in our revised manuscript. (33) Fig 2f: a legend is missing, and I can only see 4 lines and a mess of shaded area which does not allow any meaningful assessment. Please re-arrange (e.g. splitting it up to 5 single plots with same Y-axis) and it would be meaningful to use month/years for x-axis. Response: Thank you for your comment. Since this figure mainly presented the TWS trend for five hotspots, which is similar to the figure 5 (see below). Therefore, we have deleted this figure in our revised manuscript. (34) Fig 3: Labelling of Y-Axis with "Water loss" and then negative values – does it imply a water gain? Please name it more meaningful. Response: Thank you for your comment. We have rectified this mistake, and replaced "water loss" of "water storage anomaly" in our revised manuscript. (35) Fig 4: what can be seen at both axis? It seems that the months are not consecutive (If I interpret it correctly as spring season), then drawing a solid line through it is misleading. Response: Thank you for your useful comment. We have aggregated monthly data to yearly data in our revised manuscript (Figure 6). (36) Fig S1: unit for Y-Axis is missing. I suggest to use month/years instead of month numbers. Why does the time series ends âĹij at month 165 whereas the other figures are ending at month âĹij177/181? Response: Thank you for your comment. The total study period is during April 2002∼June 2017, but we use full years for comparison between 2003 and 2016, therefore the time series is during 1∼168. We have reproduced the figure by using month/year (Figure S1). (37) Fig S3: what is shown at X- and Y-Axis? Response: Thank you for your comment. We have reproduced this figure in our revised manuscript (Figure S4). (38) I have not checked if the references are listed in the reference list and vice versa, and also have not checked the reference list itself. Response: Thank you for your comment. We have carefully read the following papers, and properly cited them in our revised manuscript. References Döll, P., Müller Schmied, H., Schuh, C., Portmann, F. T., & Eicker, A. (2014). Global-scale assessment of groundwater depletion and related groundwater abstractions: Combining hydrological modeling with information from well observations and GRACE satellites. Water Resources Research, 50(7), 5698–5720.https://doi.org/10.1002/2014WR015595 Scanlon, B. R., Zhang, Z., Save, H., Sun, A. Y., Müller Schmied, H., van Beek, L. P. H., et al. (2018). Global models underestimate large decadal declining and rising water storage trends relative to GRACE satellite data. Proceedings of the National Academy of Sciences, 201704665. https://doi.org/10.1073/pnas.1704665115ȄĘ Scanlon, B. R., Zhang, Z., Rateb, A., Sun, A., Wiese, D., Save, H., Beaudoing, H., Lo, M. H., Müller Schmied, H., Döll, P., van Beek, R. Swenson, S., Lawrence, D., Croteau, M., Reedy, R. C. (2019). Tracking seasonal fluctuations in land water storage using global models and GRACE satellites. Geophysical Research Letters 46 (10), 5254-5264, 10.1029/2018GL081836 Syed T.H., Famiglietti J.S., Rodell M., Chen J., Wilson C.R. (2008). Analysis of terrestrial water storage changes from GRACE and GLDAS. Water Resour Res 44:W02433 Tangdamrongsub, N., Han, S.-C., Tian, S., Schmied, H. M., Sutanudjaja, E. H., Ran, J., & Feng, W. (2018). Evaluation of groundwater storage variations estimated from GRACE data assimilation and state-of-the-art land surface models in Australia and the North China Plain. Remote Sensing, 10(3). https://doi.org/10.3390/rs10030483. Wang, J., Song, C., Reager, J.T., Yao, F., Famiglietti, J.S., Sheng, Y., MacDonald, G.M., Brun, F., Müller Schmied, H., Marston, R.A., Wada, Y. (2018). Recent global decline in endorheic basin water storages. Nature Geoscience 11, 926-932, doi:10.1038/s41561-018-0265-7. Zhang L, et al. (2017) Validation of terrestrial water storage variations as simulated by different global numerical models with GRACE satellite observations. Hydrol Earth Syst Sci 21:821–837.   References Barnston, A. G., and Livezey, R.E.: Classification, seasonality and persistence of low-frequency atmospheric circulation patterns. Mon. Weather Rev., 115: 1083-1126, 1987. Brun, F., Berthier, E., and Wagnon, P.: A spatially resolved estimate of High Mountain Asia glacier mass balances from 2000 to 2016. Nat. Geosci., 10(9): 668-673, doi:10.1038/ngeo2999, 2017. Crétaux, J. F., Jelinski, W., and Calmant, S.: SOLS: A lake database to monitor in the Near Real Time water level and storage variations from remote sensing

[revised manuscript text omitted]

Please also note the supplement to this comment:
https://www.hydrol-earth-syst-sci-discuss.net/hess-2019-281/hess-2019-281-AC1-supplement.pdf

---

## Author Comment (AC2) · 1 Jan 2020

We would like to thank the reviewers for their professional, detailed and constructive comments, which improved our manuscript considerably. We have carefully revised the manuscript following their comments point by point. Our revisions and explanations have been inserted in blue, and all amendments are also highlighted in the version of revised manuscript. Additionally, the writing of our revised manuscript are also under carefully editing by English native speaker with specialized in hydrology. Anonymous Referee #2 The manuscript titled "Widespread decline in terrestrial water storage and its link to teleconnections across Asia and Eastern Europe" by Liu et al., has identified

an interesting research gap of analyzing the linkage between teleconnections (TCs) with terrestrial water storage (TWS) in Asia and Eastern Europe. They have utilized comprehensive set of TCs for the study. The TWS has been abstracted from GRACE observations. The TWS is partitioned using GLDAS to generate surface water (SW), soil moisture (SM) and groundwater. The TWS components are then de-seasonalized. This is followed by spatiotemporal trend analysis, comparison analysis with TCs and dissection of each TWS component's contribution to TWS. Although the manuscript embeds a promising research topic, the level of write up lags far behind the study done which in turn lags behind the research gap stated. The manuscript lacks crisp, clear messages. Most of the time this is due to poor sentence structure and grammar. The reader has to infer what the authors are trying to state or sometimes even conclude. I would not recommend to accept the manuscript in its current form and structure. I would suggest the following major revisions to the authors, if the editor decides to move the process forward. Response: Thank you very much for your thoughtful and careful comments, which have significantly improved our manuscript. We have substantially revised our manuscript point by point based on reviewers' comments in our revised manuscript. Major Comments (1) It is mentioned that the lead author wrote the manuscript with contributions from all others. However, there are significant improvements required in the sentences, paragraphs and information sequence structure. This shows the manuscript was not sufficiently revised before submission. In its current form, the manuscript doesn't facilitate an uninterrupted flow. Response: Thank you for your comment. We feel sorry for the confusion and inconvenience we have caused to you. We have carefully modified the structure, paragraph and sentence of the manuscript. (2) Tabulation of data information and a flow diagram of data processing are necessary in the section "Data". It might be a better idea to make an overall methodology flow diagram that includes this. Response: Thank you for your constructive comment. We have added a table with data information (Table 1, see attached supplement, hereafter), and we also supplemented the flow diagram of overall methodology in our revised manuscript (Figure 2, marked by the figure number in the revised

manuscript and attached supplement, hereafter). (3) There is no background on TCs and subject matter of the manuscript in regards to the study area in the section "study area". Response: Thank you for your comment. We have added the background on TCs and subject matter of our manuscript in the section of "study area" as follows. The Asian and Eastern European regions, a total of 54% of the area is arid and semiarid, are located between latitudes 6°S and 56°N and longitudes 4°E and 109°E (Figure 1a). These regions are the most densely populated regions in the world, sustaining nearly half of the global population and contain some of the largest and most intensively irrigated lands of the world (Figure 1b). The freshwater availability in these water-limit regions are essential to food and water security and hence sustainable economics. Notably, surface freshwater is critically limited in these regions (Wang, 2018). The amount of available freshwater in these regions are highly dependent on precipitation and temperature, which are influenced intensively by the Northern Hemisphere atmospheric circulation patterns and the coupled ocean-atmosphere patterns (i.e., teleconnections). The spatial explicit analysis of the impact of teleconnections on freshwater availability in these regions can be studied to provide a simple framework for understanding the complex response of freshwater availability to global climate change. (4) Although the subtopic headings of results section are clear, the content is very poorly structured and sequenced. Smooth reading flow is missing. Response: Thank you for your comment. We have substantially revised the sentences and structure of our manuscript according to the comments in our revised manuscript. (5) Although the subtopic headings of discussion section are clear, the content is very poorly structured and sequenced. For an instance, in section "4.2 Possible mechanisms of TC influence on TWS variability", only the dominant TC and their role in the regions are mentioned. However, the result from this study and discussion on "possible linkage" with these "roles" are missing. Response: Thank you for your thoughtful comment. We have reorganized the content of this section in our revised manuscript. (6) The figures and tables of the supplementary section are directly and heavily referred to in the manuscript. This needs to be ratified. The grouping of figures is also not optimal. Response: Thank you for your

comment. We have moved the heavily referred supplementary figures to main body in our revised manuscript. (7) The preparation of figures and tables is poor and needs significant revising. This includes axes labeling, color combination, color code, headers, data source declaration in the captions. Response: Thank you for your comment. We have reproduced all the figures and tables in our revised manuscript according to the useful comment. We also attached the revised figures at the end of this response. Specific Comments (in order of sections of the draft) 1 Introduction (1) Line 38: May be the gap is mainly in Asia. There are existing studies for Europe. E.g. Rakovec et al. (2016) have already analyzed the TWS anomaly using GRACE in 400 European river basins. Response: Thank you for your comment. We have carefully read this paper in the revision of the introduction section in our revised manuscript. (2) Line 42: "...are undergoing intensive..." Response: Thank you for your comment. We have revised the grammar mistake, and we also carefully check the revised manuscript. (3) Line 44: mm/y vs mm yr-1 (in abstract). Inconsistent usage of unit format. Response: Thank you for your careful comment. We have corrected the mistake in our revised manuscript. (4) Line 63-64: "...and the remainder of the TWS time series..." It's not clear for the reader what this is referring to. Response: Thank you for your comment. We have paraphrased this sentences to "we use detrended and deseasonalized TWS time series" in our revised manuscript. 2.1 Study Area (5) Figure 1a: Source/ citation is missing. Color representation for Semi-arid and dry subhumid regions don't have sufficient contrast. Response: Thank you for your comment. We have reproduced the figure 1. Panel (a) is the spatial distribution of arid and semiarid areas based on averaged aridity index during 2002-2017. The aridity index is calculated based on the ERA-Interim dataset downloaded from European Centre for Medium-Range Weather Forecasts. Panel (b) is the percentage area of irrigated land across the study area. The percentage area of irrigated land dataset is derived from Food and Agriculture Organization of the United Nations. (6) Line 72: Mount Kilimanjaro is in Africa, not the study area. Response: Thank you for your comment. We have corrected the mistake in our revised manuscript. (7) Line 73: The URL link doesn't have public access. And its

mentioning here is also not clear. Response: Thank you for your comment. We have replaced the URL link of data source citation in our revised manuscript. (8) Line 75: I couldn't grasp the usage of "...in this area". Did you mean to say "More importantly, the Asian and Eastern European regions are the most densely populated regions in the world, sustaining nearly half of the global population and contain some of the largest and most intensively irrigated lands of the world"? If so, kindly cite the source. Response: Thank you very much for your comment. We have revised the sentences according to your useful suggestions and also added the source. (9) Figure 1b: Source is missing. Response: Thank you for your comment. We have added the source of Figure 1b in Figure 1 caption as follow in our revised manuscript. The percentage area of irrigated land dataset is derived from Food and Agriculture Organization of the United Nations. (10) Line 75-77: Split long sentence to two. Also kindly include the source. Response: Thank you for your kindly comment. We have rewritten this paragraph and deleted this sentence in our revise manuscript. (11) Line 69-77: The research gap of comprehensive TWS-TC correlation would need some background on TCs in the study area. This is missing and should be discussed in more detail in this section. If the manuscript space permits, additional map/s depicting the TCs and their role in the regions would improve the clarity of the topic to the readers. Response: Thank you for the constructive comment. We have added the background on TCs and subject matter in the of "study area" section of our revised manuscript. 2.2 Data (12) Line 79-110: Tabulating the dataset information would be a more efficient way of presenting than the current form. E.g. Line 91 mentions the exact same thing mentioned in previous sentence just adding additional spatial information. This is followed by the web URL for the dataset. Tabulation of info would be the perfect solution here. Response: Thank you for your constructive comment. We have added the dataset table in our revised manuscript. (13) Line 92: Avoid using direct URLs for data reference. There are better ways to cite datasets. Response: Thank you for your comment. We have replaced the URLs of datasets citation in our revised manuscript. (14) Line 93: Bicubic interpolation doesn't preserve the mass while resampling. Mass conservative remapping is advised

(e.g. remapcon operator of CDO) Response: Thank you for your comment. We have attempted different resampling methods in revision process, i.e. nearest neighbor and bilinear interpolation. We adopted the nearest neighbor interpolation method in our revised manuscript in order to preserve the original data values. Thank you for your advice. The climate data operators (CDO) is a powerful tool in time series data set. (15) Line 94-97: This sentence has 62 words! This doesn't help readability of the manuscript. Kindly break into shorter sentences. One message per sentence. Response: Thank you for your comment. We have rewritten this paragraph and split this long sentence into short sentences in our revised manuscript. (16) Line 98: The x-axis header for Figure S1 is missing. Moreover, the x-axis is not uniform across figures S1, S2, S3, S6. Response: Thank you for your comment. We have reproduced all figures in our revised manuscript and supporting information. We have attached these figures at the end of this response. (17) Line 101: "... by deducting..." Response: Thank you for your comment. We have revised the grammar mistake in our revised manuscript, and we also carefully checked the revised manuscript. (18) Line 79-110: Apart from tabulating the dataset information, the data assimilation approach would be much easier presented as flow diagram. Response: Thank you for your constructive comment. We have provided the methodology flow diagram of the detail data processing and analysis in our revised manuscript. (19) Line 105-110: This can go into the data table. However, some elaboration of these 12 TCs in regards to the study area is missing. I would suggest to switch the position of 2.1 and 2.2. With this new sequence, the authors can provide further insight on TCs from view point of study area in the section "Study Area". Response: Thank you for your comment. We have supplemented briefly elaboration of these 12 TCs in data section of the revised manuscript as follows. The term teleconnection may refer to patterns arising from the internal variability of the atmosphere only also from the coupling between the air and the ocean. In this study, we analyze the TCs that dominate climate variability in the Northern Hemisphere, namely, Arctic Oscillation (AO), North Atlantic Oscillation (NAO), East Atlantic (EA), East Atlantic/Western Russia (EAWR), Scandinavia (SCAND), Polar/Eurasia (polarEA), West Pacific (WP),

Pacific/North America (PNA), and four important atmosphere-ocean coupled variability patterns that influence global climate, the Indian Ocean Dipole (IOD), the Atlantic Multidecadal Oscillation (AMO), the Pacific Decadal Oscillation (PDO), and ENSO (Zhu et al., 2017). The 8 first indices refer to Northern Hemisphere atmospheric circulation patterns. These 8 first indices were calculated for region 20°N- 90°N using a rotated principal component analysis (RPCA) of monthly mean standardized 500-mb height anomalies fields (Barnston and Livezey, 1987). The IOD is defined by the difference in sea surface temperature between two areas – a western pole in the western India Ocean (50∼ 70° E, 10° S∼ 10° N) and an eastern pole in the eastern Indian Ocean (90∼110° E, 10° S∼ EQ). The IOD affects the climate of Asia, and is a significant contributor to rainfall variability in this region (Saji et al., 1999). The AMO and PDO index are defined as the leading principal component of the North Atlantic Ocean (0-65°N, 80°W-0°E) and the North Pacific Ocean (poleward to 20°N) monthly sea temperature variability, respectively (Enfield, 2001; Bond, 2000). ENSO is the most important coupled ocean-atmosphere phenomenon driving global climate variability. We adopted the monthly Multivariate ENSO Index (MEI) in this study, which takes into consideration variability both in the atmosphere and in the ocean (Wolter and Timlin, 2011). 2.3 Methods 2.3.1 Time series decomposition (20) Line 113: The first sentence here is out of subject. The current subject is decomposition and this sentence is about trend analysis (belongs to section 2.3.2?) Response: Thank you for your comment. We have moved this sentence to section 2.3.2 in our revised manuscript. (21) Line 113-124: Revise the order of the sentences and info. Not in optimum order. Response: Thank you for your comment. We have revised the paragraph in our revised manuscript. (22) Line 118: STL is a robust method? Then cite the papers who have proven this method to be robust. Response: Thank you for your comment. We have added the citations in our revised manuscript. (23) Line 118: "for detecting non-linear time series in trend estimates". What do you mean? The sentence doesn't make sense to me. Response: Thank you for your comment. We have paraphrased this sentence in our revised manuscript. (24) Line 123-124: Refer to previous publication of this journal to

understand how to cite in such situation. Response: Thank you for your comment. We have modified the citation in our revised manuscript. 2.3.2 Theil-Sen trend analysis (25) Line 126: "..the linear trend of .... and precipitation for the ..." Response: Thank you for your comment. We have revised this sentence based on the useful comment in our revised manuscript. (26) Line 126: Trend analysis on the deseasonalized time series? or the residuals? I am guessing the first sentence of section 2.3.1 belongs here [??] Response: Thank you for your comment. Trend analysis in this study is based on the deseasonalized time series. We have revised this sentence in our revised manuscript. (27) Line 126-127: Break the long sentence into two. Move the second part to the end of the section. In this way flow of read regarding Theil-Sen trend analysis is maintained. Response: Thank you for your comment. We have modified this section based on the useful comment in our revised manuscript. (28) Line 128: "... of the Theil-Sen trend analysis is ..." Response: Thank you for your comment. We have added "analysis" in our revised manuscript. (29) Line 130: Remove "non-robust". And cite the literature proving this statement. Response: Thank you for your comment. We have revised this sentence and added citation in our revised manuscript. (30) Line 130: "The TWS trend, ß, for a ..." Response: Thank you for your comment. We have revised this sentence based on your comment in our revised manuscript. 2.3.3 Cross-correlation analysis (31) Line 136: TWS or TWS residual? Response: Thank you for your comment. We have replace TWS of TWS residual, and reorganized this section in our revised manuscript. (32) Line 139: Move the "," to the end of the equation. At the moment its on the denominator. Response: Thank you for your comment. We have revised this mistake in our revised manuscript. (33) Line 139: What is the meaning of the symbol Tau here? Is it the lag? Response: Thank you for your comment. The symbol Tau is the time lag, and we have added the explanation in our revised manuscript. (34) Line 140-141: "auto-covariace"? Response: Thank you for your comment. We have revised this word in our revised manuscript. (35) Line 142: Why between 0 and 24? Reason, in brief, required. Moreover, can cross correlation have value greater than 1? Response: Thank you for your comment. The value of cross correlation coefficient lies between

-1 and +1. We have corrected the statement in our revised manuscript. Additionally, the multiple TCs could reflect different influence of atmosphere and ocean variability on TWS from short-term to long-term. For example, the impact of AO and NAO have a relatively high-frequency variability on TWS. Therefore, we adopt the lag of 0-24 month in current study in order to address the different time scale responses of TCs on TWS. (36) Line 143-146: Break the sentence to get one message per sentence. Response: Thank you for the comment. We have broken this long sentence into short sentences in our revised manuscript. 3 Results 3.1 Spatiotemporal changes in TWS (37) Line 149: Mention that both JPL-M and CSR-M showed similar spatiotemporal pattern to begin with. Then let the reader know that the values will be referred to JPL-M. Response: Thank you for your comment. We have supplemented these sentences in first of section 3.1 in our revised manuscript. (38) Line 152: The 5 hotspots are clearly shown in Figure 2a. This should be included in the reference illustrations. Response: Thank you for your comment. We have added the reference illustration in our revised manuscript. (39) Line 152: Figure 2f doesn't have color code. Unit of time is missing. Time axis with years as axis tick labels would enhance clarity. Response: Thank you for your comment. Since this figure mainly presented the TWS trend for five hotspots, which is similar to the figure 5. Therefore, we have deleted this figure in our revised manuscript. (40) Line 153: Figure S6 doesn't have color code. Response: Thank you for your comment. We have reproduced the figure in our revised manuscript. (41) Line 154: Precipitation is in figure 2b, not 2c. Response: Thank you for your comment. We have corrected this mistake in our revised manuscript. (42) Line 157: Precipitation is in figure 2b, not 2c. Response: Thank you for your comment. We have corrected this mistake in our revised manuscript. (43) Line 158: How is this "within" Rodell's findings? The estimates are completely contrasting each other. Plus, the reasoning is not convincing and/or explained properly. Response: Thank you for your comment. We have rewritten this sentence in our revised manuscript. (44) Line 162: -73.2 mm y-1 is out of range for figure 2a. Response: Thank you for your comment. The trend in Figure 2a (Figure 3a in revised manuscript) is the changes in terrestrial water storage

over the study area (equivalent water depth), whereas the trend of -73.2 mm y-1 is the water level variation of the Caspian Sea. 3.2 Influence of TC indices on TWS variability (45) Line 174: Which section of the multi-plot Figure 2? Revise the header and legend header of figure S5 to something more explicit. In caption of Figure S5, replace "phase shift" by "lag". Usage of same terminology maintains the flow of reading. And what are the symbol alpha and "n" in the caption? Response: Thank you for your comment. We have reproduced these figures, and added clarifications in figure captions. (46) Line 175: "... and NAO have a significant area of influence on TWS variability." Response: Thank you for your comment. We have revised the sentence according to you suggestion in our revised manuscript. (47) Line 181: Resolve the structural error. Two sentences or one? Response: Thank you for your comment. We have revised the structural error in our revised manuscript. (48) Line 189: "Proportions of time .... Figure 2d". This sentence should be the starting sentence of this paragraph. Response: Thank you for your comment. We have revised this sentence according to you useful suggestion in our revised manuscript. (49) Line 190: Tibetan plateau and Mongolia have more pixels of longer lags than SE Asia. Response: Thank you for your comment. We have rewritten this sentence in our revised manuscript. 3.3 Contributions of water storage components to TWS (50) Line 194: The first sentence is concluding findings. Thus, it is more suitable to be placed towards the end of the section. Response: Thank you for your comment. We have moved the first sentence to the end of the section in our revised manuscript. (51) Line 195: The hotspots have been well established in the manuscript and doesn't need the reference to Figure 2a every time. Response: Thank you for your comment. We have deleted the reference of hotspots in our revised manuscript. (52) Line 195: "Groundwater depletion dominates the contribution to TWS loss in region 1..." Response: Thank you for your comment. We have revised this sentence in our revised manuscript. (53) Line 197: "Similar results were observed in northwest..." Response: Thank you for your comment. We have revised this sentence in our revised manuscript. (54) Line 210: The term "sea level" could be confused with the world mean sea level. Suggested rephrasing: "...Caspian Sea with a

decrease in its water level elevation by -73.2 mm y-1..." Response: Thank you for your comment. We have rephrased this sentence according to your useful suggestion in our revised manuscript. 3.4 Divergent response of water storage components to TCs (55) Line 217: The first sentence is concluding findings. Thus, it is more suitable to be placed towards the end of the section. Response: Thank you for your comment. We have moved the first sentence to the end of the section in our revised manuscript. (56) Line 222: "...is a synthesis signal i.e. its trend ...". "... different ways. Furthermore, the groundwater ..." Response: Thank you for your comment. We have revised this sentence in our revised manuscript. (57) Line 223: "...which indicates lower correlation..." Response: Thank you for your comment. We have replaced the word "less" of "lower" in this sentence in our revised manuscript. (58) Line 225: Reference to a figure or table missing Response: Thank you for your comment. We have added the reference in this sentence in our revised manuscript. 4 Discussion 4.1 Comparison of our results to previous studies (59) Line 235-257: Please be clear about which region are you talking about. If you start with "Region 1 shows ..." then its clear that the information corresponding to that region (1, 3 and 4) else its hard to follow (regions 2 and 4). Response: Thank you for your comment. We have rewritten this section in our revised manuscript as follows. We investigate the spatiotemporal trend of TWS and its components over Asia and Eastern Europe region during 2002-2017. The spatial pattern and trend of TWS over the study area are consistent with those of previous studies (Humphrey et al., 2016; Scanlon et al., 2016). Our estimate trend of TWS in region 1 is within that of previous studies in this region (22±3 mm yr-1 during 2003-2010) (Feng et al., 2013). Due to a long-term warm and dry climate and intensive anthropogenic activities (agriculture, industry, and urbanization), the groundwater in region 1 has been overexploited since the 1970s, and more than 70% of the groundwater exploitation is used for regional irrigation (Wang et al., 2007). The rate of groundwater loss was also reported by a previous study in region 3 (approximately 40±10 mm yr-1 from August 2002 to October 2008) (Rodell et al., 2009). As Indian agriculture leads the world in total irrigated land by consuming ∼85% of the utilizable water resources

(Panda et al., 2016; Salmon et al., 2015), a concluding consensus has been reached that the dramatic decline in TWS is mainly due to the overexploitation (extraction exceeding recharge) of groundwater for irrigation (Shamsudduha et al., 2019). Although precipitation in region 3 shows an increasing trend during the GRACE period, the rapid depletion of TWS in northwest India induced by unsustainable consumption of groundwater for irrigation and other anthropogenic uses has attracted worldwide attention because it is a major threat to India's sustainability (Panda et al., 2016; Rodell et al., 2009). Region 4 in our study is also heavily irrigated (Figure 1), so intensive irrigation is likely to induce groundwater decline. The increase in SW induced by melt water from mountains (Brun et al., 2017) was offset by the decrease in soil water that may be related to the decrease in precipitation (Figure 2b). For region 2, the rapidly melt of the glaciers of Tien Shan Mountain accelerate an increase in the loss of water resources, since the glacial meltwater will provide additional water that was lost to rivers or evaporation (Jacob et al., 2012). The negative trend in TWS indicates that water demand is larger than supply in region 2, which can be attributed to both continuous withdrawal of groundwater and extensive evaporation in the endorheic basin (Rodell et al., 2018). However, the increase in precipitation is expected to offset a certain portion of water losses in region 2. Previous studies documented that the widespread decline in TWS in region 5 is also attributed to the overreliance on groundwater for domestic and agricultural needs due to human-made dams in addition to the sharply surface water loss (Joodaki et al., 2014; Rodell et al., 2018; Voss et al., 2013), these reports are consistent with our results. (60) Line 258-260: Break the sentence to get one message per sentence. Response: Thank you for your comment. We have reorganized this paragraph in our revised manuscript. (61) Line 260-263: Break the sentence to get one message per sentence. Response: Thank you for your comment. We have reorganized this paragraph in our revised manuscript. (62) Line 263-264: Revise the sentence for clarity. The concluding sentence should be clear. Response: Thank you for your comment. We have revised this sentence for clarity in our revised manuscript. 4.2 Possible mechanisms of TC influence on TWS variability (63) Section 4.2: This

is probably the most interesting subtopic of the paper. The writing style should have followed this pattern: 1) result observation at each hotspot, 2) literature on the dominant TC for the hotspot, 3) linking "possible mechanism" between the literature and the results. Currently the section is filled with only 2. There is no linking going on. Response: Thank you for your constructive comment. We have reorganized section 4.2 according to your comment in our revised manuscript. 4.3 Implications for future hydrological studies (64) Line 288: "... could explain the variability in TWS in most of the remote and ..." Response: Thank you for your comment. We have revised this sentence in our revised manuscript. (65) Line 290: "... variability interacts with human..." Response: Thank you for your comment. We have revised this sentence in our revised manuscript. (66) Line 297: "claim" is a very strong word. Moreover, it doesn't make sense especially as the paper hasn't included any prediction or scenario analysis of droughts and heatwaves. Response: Thank you for your comment. We have replaced the word "claim" of "infer" in our revised manuscript. (67) Line 300-314: Usage of "First, Second, Third" in paragraph structure requires different approach of writing. Instead, bullet style enumeration of the three recommendations would suit the current sequence and structure of writing i.e. start first bullet with "Withdrawal of ...", and so on. Response: Thank you for your comment. We have revised the usage of "First, Second, Third" by using bullet style enumeration in our revised manuscript. 5 Conclusions (68) Line 321: "...component vary from region to region. The ..." Response: Thank you for your comment. We have revised this sentence in our revised manuscript. (69) Line 323: "...and regions. This highlights the importance ...." Response: Thank you for your comment. We have revised this sentence in our revised manuscript. References: RAKOVEC, O., KUMAR, R., MAI, J., CUNTZ, M., THOBER, S., ZINK, M., ATTINGER, S., SCHÄFER, D., SCHRÖN, M. and SAMANIEGO, L.: Multiscale and Multivariate Evaluation of Water Fluxes and States over European River Basins, J. Hydrometeorol., 17, 287–307, doi:10.1175/JHM-D-15-0054.1, 2016. Response: Thank you very much for all the comments.   References Barnston, A. G., and Livezey, R.E.: Classification, seasonality and persistence of low-frequency atmospheric circulation patterns. Mon. Weather Rev., 115: 1083-1126, 1987. Brun, F., Berthier, E., and Wagnon, P.: A spatially resolved estimate of High Mountain Asia glacier mass balances from 2000 to 2016. Nat. Geosci., 10(9): 668-673, doi:10.1038/ngeo2999, 2017. Crétaux, J. F., Jelinski, W., and Calmant, S.: SOLS: A lake database to monitor in the Near Real Time water level and storage variations from remote sensing data. Adv. Space Res., 47(9): 1497-1507, doi.org/10.1016/j.asr.2011.01.004, 2011. Enfield, D.B., Mestas-Nunez, A. M., and Trimble P. J.: The Atlantic multidecadal oscillation rainfall and river flows in the continental and its relation U.S. Geophys Res Lett, 28(10): 2077-2080, 2001. Feng, W., Zhong, M., and Lemoine, J. M.: Evaluation of groundwater depletion in North China using the Gravity Recovery and Climate Experiment (GRACE) data and ground-based measurements. Water Resour. Res., 49(4): 2110-2118, doi: 10.1002/wrcr.20192, 2013. Humphrey, V., Gudmundsson, L., and Seneviratne, S. I.: Assessing Global Water Storage Variability from GRACE: Trends, Seasonal Cycle, Subseasonal Anomalies and Extremes. Surv. Geophys., 37(2): 357-395, doi: 10.1007/s10712-016-9367-1, 2016. Jacob, T., Wahr, J., and Pfeffer, W.: Recent contributions of glaciers and ice caps to sea level rise. Nature, 482: 514-518, doi: 10.1038/nature10847, 2012. Joodaki, G., Wahr, J., and Swenson, S.: Estimating the human contribution to groundwater depletion in the Middle East, from GRACE data, land surface models, and well observations. Water Resour. Res., 50(3): 2679-2692, doi: 10.1002/2013wr014633, 2014. Landerer, F. W., and Swenson, S. C.: Accuracy of scaled GRACE terrestrial water storage estimates. Water Resour Res, 48: w04531, doi:10.1029/2011WR011453, 2012. Panda, D. K. and Wahr, J.: Spatiotemporal evolution of water storage changes in India from the updated GRACE-derived gravity records. Water Resour. Res., 52(1): 135-149, doi: 10.1002/2015wr017797, 2016. Rodell, M., Famiglietti, J. S., and Wiese, D. N.: Emerging trends in global freshwater availability. Nature, 557: 651-659, doi: 10.1038/s41586-018-0123-1, 2018. Rodell, M., Houser, P. R., and Jambor, U.: The Global Land Data Assimilation System. B. Am. Meteorol. Soc., 85(3): 381-394, doi: 10.1175/bams-85-3-381, 2004. Rodell, M., Velicogna, I., and Famiglietti, J. S.: Satellite-based estimates of groundwater deple-

[revised manuscript text omitted]

Please also note the supplement to this comment:
https://www.hydrol-earth-syst-sci-discuss.net/hess-2019-281/hess-2019-281-AC2-supplement.pdf

———————————————————

---

## Referee Report (RR1)

Reviewer Comments: hess-2019-281

The authors have addressed the comments in the manuscript. I believe the presentation quality of the work (figures, tables, information structure) has improved. They have also worked a bit on improving the discussion section whiich was important. Accepted subject to technical (remaining) corrections.

Technical Corrections:

Line 75: The reference is not listed in Bibliography

Line 143-144: "For detailed principles and applications of STL, readers are encouraged to refer to Cleveland et al. (1990) and Bergmann et al. (2012)."

Previous comment 34: Not yet addressed in the manuscript.

---

## Author Response (AR2)

**Response to referees' comments (hess-2019-281)**

We would like to thank the reviewers for their professional, detailed and constructive comments, which improved our manuscript considerably. We have carefully revised the manuscript following their comments point by point. Our revisions and explanations have been inserted in blue, and all amendments are also highlighted in the version of revised manuscript.

**Anonymous Referee #2**

The authors have addressed the comments in the manuscript. I believe the presentation quality of the work (figures, tables, and information structure) has improved. They have also worked a bit on improving the discussion section which was important. Accepted subject to technical (remaining) corrections.

(1) Line 75: The reference is not listed in Bibliography

Response: Thank you for the comment. We have added the reference in Bibliography as follows. *Center for International Earth Science Information Network - CIESIN - Columbia University. Gridded Population of the World, Version 4 (GPWv4): Administrative Unit Center Points with Population Estimates, Revision 11. Palisades, NY: NASA Socioeconomic Data and Applications Center (SEDAC), 2018.*

(2) Line 143-144: "For detailed principles and applications of STL, readers are encouraged to refer to Cleveland et al. (1990) and Bergmann et al. (2012)."

Response: Thank you for the carefully comment. We have revised the format of these two references in our revised manuscript.

(3) Previous comment 34: Not yet addressed in the manuscript

Response: Thank you for your comment. We have added "τ is the time lag" in section 2.3.3 of our revised manuscript.

**Response to referees' comments (hess-2019-281)**

We would like to thank the reviewers for their professional, detailed and constructive comments, which improved our manuscript considerably. We have carefully revised the manuscript following their comments point by point. Our revisions and explanations have been inserted in blue, and all amendments are also highlighted in the version of revised manuscript.

**Anonymous Referee #3**

Manuscript of Liu et al. (2020) evaluates the effect of decline in terrestrial water storage across Europe and Asia, derived from GRACE products are complemented with GLDAS LSM simulations. Trends in TWS dynamics are discussed and related to major global teleconnections with associated time lags. I review this manuscript for the first time. I see that authors improved a lot their narration since the original submission. I find this topic interesting and relevant for HESS. The manuscript is clearly written. Below are several minor comments, which should be considered.

(1) Section 3.1: comparison of TWS anomaly with precipitation is useful, but not complete. How about the contribution of global temperature rise and increase evaporative rates? Should be mentioned in this section to complement CRU precipitations.

**Response: Thank you for the useful comment. We have produced the maps of changes in temperature and evapotranspiration across study area during 2002-2017, and added analysis in section 3.1.**

[Figure]

[Figure]

**Figure: Spatiotemporal changes in temperature as obtained from CRU (left) and evaporation as obtained from ERA5 (right) across the Asian and Eastern European regions during 2002–2017.**

(2) In your study you evaluate multiple teleconnections. Their interdependencies could have been further decomposed to detect and quantify causal associations among them. I feel it is not necessary to fully implement, but you might have a short discussion on the method presented by Runge et al. 2019 (doi: 10.1126/sciadv.aau4996) and references cited therein in the discussion.

**Response: Thank you for the useful comment. We have carefully read this paper, and added it in discussion section in our revised manuscript.**

(3) Here are some relevant papers fitting in your manuscript: Yi et al. (2014, doi: 10.1002/2013JB010860) discussing glacier changes in Asia with GRACE and TCs, and Forootan (2019, doi: 10.1016/j.scitotenv.2018.09.231) discussing

IOD and NAO having regional influence on the evolution of hydrological droughts.

**Response: Thank you for the comment. We have carefully read these two papers, and also cited in our revised**

**manuscript.**

(4) Section 3.3 needs to start with statement these are based on the Noah LSM.

**Response: Thank you for the comment. We have introduced the Noah LSM in data section 2.2, and the results in**

**section 3.3 are based on GRACE-based solutions, the Noah LSM data, and other lakes and glacial data. Therefore,**

**I am not sure if it is suitable to start with statement that these are based on the Noah LSM.**

(5) Line 108: Figure S1 is not much useful, could be valuable to decompose into several sub-regions (e.g. the same 5

discussed later, Table 1). It does not have much meaning showing it across entire 1.5continents.

**Response: Thank you for the comment. Many existing studies have compared the differences between GRACE-**

**based and GLDAS-based TWS in different scales. Our previous work (doi.org/10.1016/j.scitotenv.2018.03.292)**

**also compared the difference in the Yangtze River Basin. Therefore, in this study, we only compared the difference**

**across the whole study area. We will further comparison the differences between GRACE-based TWS and**

**GLDAS-based TWS in multiple scales in our future work.**

**Additional comments:**

(1) line 24: cycles => cycle

**Response: Thank you for the comment. We have replaced cycles with cycle in our revised manuscript.**

(2) Line 35-37 (split references for the three scales)

**Response: Thank you for the comment. We have separated the references into three scales in our revised**

**manuscript.**

(3) Line 43: remove "fortunately" or replace by more neutral term.

**Response: Thank you for the comment. We have deleted this word in our revised manuscript.**

(4) Line 46: "threats" replace by "exploitation"

**Response: Thank you for the comment. We have replaced "threats" with "exploitation" in our revised manuscript.**

(5) Line 50: challenges => challenge

**Response: Thank you for the comment. We have revised this word in our revised manuscript.**

(6) Line 56: "land" => could you tell which part of the Globe?

**Response: Thank you for the comment. We have revised this statement in our revised manuscript.**

(7) Line 67: specify LSM name here

**Response: Thank you for the comment. We have added the LSM name in our revised manuscript.**

(8) Line 103: thirty-six parameters => "36 variables"

**Response: Thank you for the comment. We have revised this statement in our revised manuscript.**

(9) Line 161-162: remove underscore from sigma's and x's names.

**Response: Thank you for the comment. We have deleted all underscore in sigma's and x's names in our revised**

**manuscript.**

(10) Line 172-173: rephrase sentence like "Both GRACE-based solutions (JPL-M and CRR-M) show …"

**Response: Thank you for the comment. We have rephrased this sentence in our revised manuscript.**

(11) Line 179 and 181: figure "2b" should be "3b"?

**Response: Thank you for the comment. We have revised this mistake in our revised manuscript.**

(12) Line 332: remove both citations from here.

**Response: Thank you for the comment. We have remove these two citations in this sentence in our revised**

**manuscript.**

(13) Figure 4a: include class break at - 0.15 and +0.15, which corresponds to your significance level limit.

**Response: Thank you for the comment. We have reproduced figure 4a, and added the break at -0.15 and +0.15 in**

**our revised manuscript.**

(14) Figure 5: caption needs to mention values originate from LSM.

[revised manuscript text omitted]